# Subcellular Stress Markers in Epithelial Ovarian Cancer

**DOI:** 10.3390/ijms27010342

**Published:** 2025-12-28

**Authors:** Edina Amalia Wappler-Guzzetta, Eva Margittai, Krisztina Veszelyi, Shanel Pickard, Caroline Merwin, Attila Molvarec, Ibolya Czegle

**Affiliations:** 1Division of Transfusion Medicine, Department of Laboratory Medicine, University of California, San Francisco, CA 94143, USA; wapplerg@bu.edu; 2Institute of Translational Medicine, Semmelweis University, 1085 Budapest, Hungary; margittai.eva@semmelweis.hu (E.M.); veszelyi.krisztina@semmelweis.hu (K.V.); 3Department of Laboratory Medicine, University of California, San Francisco, CA 94143, USA; shanel.pickard@ucsf.edu; 4Delgado Community College, New Orleans, LA 70119, USA; caroline.merwin@dcc.edu; 5Department of Obstetrics and Gynecology, Semmelweis University, 1085 Budapest, Hungary; molvarec.attila@semmelweis.hu; 6Department of Internal Medicine and Hematology, Semmelweis University, 1085 Budapest, Hungary

**Keywords:** epithelial ovarian cancer, mitochondrial stress, endoplasmic reticulum stress

## Abstract

Epithelial ovarian cancer is one of the most lethal gynecological malignancies worldwide. Its development strongly depends on several genetic and environmental factors, with metabolic components and cellular redox homeostasis alterations playing a significant a role in its development and disease progression. In this review, we summarize the contribution of mitochondrial and endoplasmic reticulum (ER) stress in the pathogenesis of epithelial ovarian cancer along with their role as potential biomarkers and therapeutic targets, including proteins of glucose metabolism, mitochondrial fission and fusion, mitophagy, membrane-associated ring-CH-type finger 5 (MARCH5), A-kinase anchoring proteins (AKAPs), proteins regulating mitochondrial Ca^2+^ homeostasis, mitochondrial unfolded protein response (UPRmt) proteins, activating transcription factors (ATFs), CCAAT enhancer binding protein (C/EBP) homologous protein (CHOP), ‘mitokines’, GRP75, and GRP78. Although many of these potential targets are in preclinical phase, they have a high potential to become valuable alternative or additive treatments for epithelial ovarian cancers.

## 1. General Aspects of Ovarian Cancer

Epithelial ovarian cancer (EOC) represents a heterogeneous spectrum of disease entities at the clinical, pathological, and molecular level. Ovarian cancer is the second most lethal gynecological malignancy [1]. Infertility or nulliparity, estrogen treatment, and obesity are risk factors for the development of EOC, and they could account for the increasing incidence of the disease in developed countries [2,3].

### 1.1. Histological Subtypes

EOC represents the majority (approximately 90%) of ovarian malignancies. The 2020 World Health Organization (WHO) classification based on histopathology, immunohistochemistry (IHC) and molecular analysis recognizes at least four distinct subtypes of EOC: serous carcinomas [high-grade serous carcinoma (HGSC; ~70%), and low-grade serous carcinoma (LGSC; ~5%)], endometrioid ovarian cancer (EC; ~10%), clear cell carcinoma (CCC; ~6–10%), and mucinous carcinoma (MC; ~3–4%), along with other rare entities, including mesonephric-like carcinoma, mixed-cell tumor, malignant Brenner tumor, carcinosarcoma, and undifferentiated carcinoma. Each subtype represents a distinct disease entity, often with a different site of origin, pathogenesis, clinical features, and prognosis. In addition, HGSC and CCC, for example, have distinct metabolic phenotypes, which further distinguish these tumors. The complexity of subclassification and its effect on personalized treatment choice underline the importance of histological tumor typing. IHC staining patterns and molecular features of the different subtypes are summarized in Table 1. Certain genomic and molecular alterations, such as *BRCA1/2*-mutation (also known as *BRCA1/2*-mut) or other homologous recombination deficiency (HRD) are helpful in predicting the effectiveness of targeted therapy with poly(ADP-ribose)polymerase inhibitors (PARPis) in high-grade tumors [4].

### 1.2. Therapeutic Considerations: Chemotherapy, Antiangiogenic Treatment, and Targeted Therapies

The backbone of ovarian cancer treatment is surgical intervention with conventional chemotherapy (ChT) (mainly platinum-based drugs), often combined with antiangiogenic drugs and other targeted therapies, such as PARP inhibitors. In this article, we only review non-surgical treatment options, given the focus on mitochondrial and ER stress in endothelial ovarian cancer.

In the early stages of the EOC, adjuvant platinum-based ChT significantly prolongs patient survival [14,15] after surgery. In addition, bevacizumab, a monoclonal vascular endothelial growth factor (VEGF) inhibitor, has shown survival benefit in clinical trials as a neoadjuvant treatment [16,17].

In advanced stages of ovarian cancer, systemic ChT, using carboplatin and paclitaxel, following cytoreductive surgery is recommended for all patients. Additionally, consideration should be given to the inclusion of antiangiogenic and maintenance ChT [18]. Two large randomized controlled trials (RCTs) showed that the addition of bevacizumab to paclitaxel and carboplatin in first-line therapy, followed by bevacizumab as maintenance therapy, resulted in a statistically significant increase in survival compared to chemotherapy alone [19]. Up to 50% of HGSCs are HRD positive, including ~15–20% of germline *BRCA1/2*-mut cases. Additionally, somatic *BRCA1/2*-mut, *BRCA1* promoter epigenetic silencing via hypermethylation, and deficiencies of other proteins in the DNA double-strand break homologous recombination repair contribute to the remainder of the positive tests [20]. HRD positivity with or without *BRCA1/2*-mut is a well-established predictive factor of the magnitude of response to PARPis. The incorporation of PARPis in the maintenance therapy after first-line ChT has led to a new era in the management of advanced HGSC/high-grade EC, with unprecedented benefit of different PARPis in clinical trials for patients with *BRCA1/2*-mut or *BRCA1/2*-wild type (wt)/but HRD-positive tumors [21,22,23,24]. Future therapeutic directions may include the combination of a PARPi with other molecules, targeting cellular metabolism to enhance the effectiveness of PARPi therapy in OC.

Up to 70% of patients with stage III-IV high-grade ovarian cancer will relapse within three years. Relapse rates for early-stage ovarian cancer are much lower. Systemic therapy of recurrent disease is based on platinum-containing or non-platinum-containing regimens. The treatment decision is based on whether the disease is platinum sensitive or resistant, with the definition of platinum sensitivity being a six-month relapse-free and ChT-free interval from the last use of platinum (TFIp). On the other hand, platinum resistance is defined as the development of relapsed disease within the first six months after completing the first line of platinum-containing ChT. Relapsed patients with platinum-sensitive tumors should be considered to receive platinum-based therapy if platinum is not contraindicated, and if there is a reasonable benefit from the platinum rechallenge (no progression during platinum-based therapy or shortly thereafter). Currently there are no molecular markers to predict efficacy of platinum rechallenge [22].

### 1.3. Antiangiogenic Therapy and PARP Inhibitors in Recurrent Disease

Bevacizumab is approved in combination with a platinum-based therapy and subsequent maintenance therapy for patients with a treatment-free interval of >6 months. Bevacizumab plus platinum (either paclitaxel or gemcitabine, followed by bevacizumab maintenance) leads to a significant benefit in objective response rate and survival [25]. Three PARPis (olaparib, niraparib, and rucaparib) are currently approved for maintenance therapy in patients with high-grade tubo-ovarian carcinoma, who has a good response to platinum rechallenge. Based on data from clinical trials, they can be used irrespective of the tumor *BRCA1/2*-mut or HRD status. Both antiangiogenic therapy and PARPis should be continued until disease progression [22].

### 1.4. Other Targeted Therapies

Other targeted therapies (e.g., EGFR inhibitors, anti-HER2 drugs, due to lack of molecular target) fail to show any survival benefit, therefore not detailed in this article. Regarding immune checkpoint inhibitors, Won-Hee Yoon et al. reviewed their role in advanced stage EOC. Numerous clinical trials investigated the use of immune checkpoint inhibitors as single-agent, dual-agent therapy, as well as in combination with other antiangiogenic agents. Although the promising preclinical data showed a possible synergistic effect between antiangiogenic treatment and immune checkpoint inhibitors, clinical trial data showed disappointing data regarding either survival outcomes or an intolerable side-effect profile, therefore these drugs have not been incorporated into routine clinical practice [26].

## 2. Mitochondrial Stress Markers

Mitochondria have a versatile function in cellular homeostasis under normal and pathological conditions. They regulate multiple key cellular processes, such as cellular metabolism, reactive oxygen species (ROS) production, calcium (Ca^2+^) signaling, apoptosis, and phospholipid and heme synthesis [27,28,29]. It is therefore not surprising that they are key players during carcinogenesis, tumor cell survival, and in the development of therapy resistance. Metabolic changes, alterations in mitochondrial mass and mitochondrial DNA (mtDNA), mitochondrial dynamics (fission/fusion and mitophagy), and other mitochondria-associated changes are discussed in this section. Their alterations in OC are discussed later, in Section 3.

### 2.1. Metabolic Changes

Cancer cells require a highly adaptive metabolism for their survival, which is considered one of the mitochondrial stress markers [27]. Understandably, mitochondrial adenosine triphosphate (ATP) production via oxidative phosphorylation (OXPHOS) is often increased in cancer cells along with increased ROS production. Cancers that are not OXPHOS-dependent increase their ATP production via enhanced anaerobic glycolysis, also known as the Warburg effect. Those, however, typically still have ongoing ATP production from OXPHOS. Mitochondrial pyruvate carriers (MPCs) play an important role in fueling Krebs (also known as TCA) cycle, transferring pyruvate from the cytosol to the mitochondrial matrix, where it forms acetyl-CoA by decarboxylation by pyruvate dehydrogenase (PDH), or oxaloacetate by carboxylation by pyruvate carboxylase (PC) [30]. MPCs work as a heterodimer of MPC1 and MPC2, forming an MPC complex, formed by their transmembrane helices [30] (Figure 1). OC-associated changes are described in Section 3.

### 2.2. Mitochondrial DNA (mtDNA) and Mitochondrial Mass

The circular mtDNA encodes 13 essential mitochondrial proteins out of the over 1000 proteins that comprise the mitochondria, with the rest encoded by the nuclear DNA [31,32].

To increase mtDNA, and subsequently mitochondrial mass, mtDNA replication must take place. The mtDNA-directed RNA polymerase enzyme, encoded by *POLRMT* gene, provides RNA primers for replication initiation of the mitochondrial genome. Afterward, DNA polymerase γ (POLγ) continues the DNA synthesis with the help of TWINKLE DNA helicase, which unwinds the mtDNA for replication [33]. During this replication process, the strand that does not undergo DNA synthesis at that time is stabilized by a protein called the mitochondrial single-stranded DNA-binding protein (mtSBB) [33]. When replication is completed on both strands, RNAse H1 and mitochondrial genome maintenance exonuclease 1 (MGME1) assist to terminate the process by removing the primers, with topoisomerase 3A (TOP3A) separating the two newly formed mtDNAs [33] (Figure 2).

Following mtDNA replication, increased mtDNA and nuclear DNA transcription and translation of mitochondrial proteins occur when cells need to increase their mitochondrial mass. Interestingly, transcription initiation also involves the mitochondrial RNA polymerase (POLRMT) along with mitochondrial transcription factor A (TFAM) and mitochondrial transcription factor B1/2 (TFB1M/2M). Transcription elongation factor (TEFM) is then involved in the elongation phase of the transcription, and the mitochondrial transcription termination factor 1 (mTERF1), terminating the transcription [27]. Certain nuclear transcription factors, such as c-Jun, Jun-D, and CEBPb can localize to the mtDNA to regulate transcription activity, adjusting the cells’ metabolic needs [27,34]. Furthermore, many mitochondrial transcription coactivators exist, including the peroxisome proliferator-activated receptor gamma coactivator-1 (PGC-1α), which are crucial for mitochondrial biogenesis [27,35]. On the other hand, transcription inhibition was described with 7S RNA, which specifically inhibits POLRMT activity [36]. For more details on their alterations in OC, see Section 3.

### 2.3. Mitochondrial Open Reading Frame of the 12S rRNA-c (MOTS-c)

Mitochondrial short open frames (sORFs) are short mtDNA sequences that once were thought to be ‘junk’ that are translated to small, functional peptides, called the mitochondrial-derived peptides (MDPs), which regulate various signaling and metabolic protein expression [37,38,39]. MDPs include MOTS-c, humanin, SHLP2, and SHLP6, which have various functions in different tissue and under different circumstances [38]. Interestingly, metabolic stress has been described to initiate AMPK-dependent MOTS-c translocation to the nucleus, where it alters gene expression to enhance metabolic stress resistance [37]—a key factor in cancer cell survival. See more details about sORFs/MOT-c in OC in Section 3.1.5.

### 2.4. Mitochondrial Fission

Mitochondrial fission/fusion and mitophagy maintain the required number of healthy mitochondria for proper cellular metabolism in both normal and cancerous cells. Furthermore, these processes are involved in apoptosis regulation. Mitochondrial fission increases the number of mitochondria, which is a hallmark of most cancer cells. Excessive fission, however, is associated with apoptosis induction. For more details of mitochondrial fission and fusion in OC, see Section 3.3, with preclinical and human data for OC found in Section 3.3.

#### 2.4.1. Dynamin-Related Protein 1 (Drp1)

The main fission regulator, Drp1 is located in the cytoplasm while inactive and recruited to the mitochondrial outer membrane (MOM) at the time of fission. Once Drp1 is on the MOM, fission occurs when Drp1 molecules, which are GTPases, are assembled into helical rings around the mitochondria, causing membrane constriction [40].

#### 2.4.2. Adaptor Proteins

Mitochondrial fission factor (Mff) and mitochondrial fission 1 protein (Fis1). Adaptor proteins located on the MOM, such as Mff, Fis1, and mitochondrial dynamics proteins of 49 kDa and 51 kDa (MiD49 and MiD51, respectively) [41,42], facilitate Drp1 recruitment, translocation, and assembly. Both Fis1 and Mff are located in the MOM, and act as an anchor to Drp1. While Mff can bind and activate Drp1 to induce fission under physiological conditions [43], Fis1 may only play an important role in stress-induced mitochondrial fission, such as in response to oxidative stress [44]. Mff is also important in regulating cellular metabolism, as it can be activated by the energy-sensing adenosine monophosphate (AMP)-activated protein kinase (AMPK) under low-nutrient conditions, increasing mitochondrial fission to enhance ATP production [45]. Furthermore, there is evidence that Fis1 can induce mitochondrial fission by inhibiting mitochondrial fusion in human cells [46].

#### 2.4.3. Adaptor Proteins

MiD49 and MiD51. Drp1 adaptor proteins MiD49 and MiD51 have very similar structures to each other, and they can bind ER besides Drp1 in the same region, making them an important part of ER-mitochondria connection and signaling [47,48]. There is, however, a significant difference between the two: while MiD51 can bind ADP/GDP, MiD49 is incapable of ADP/GDP binding due to its smaller nucleotide-binding pocket [49]. In functional studies, ADP was found to be an essential cofactor for MiD51, facilitating mitochondrial fission. Low-rate MiD51-Drp1 binding and mitochondrial fission, however, still occurs with mutant MiD51 that has no ADP/GDP binding ability [50]. Also, MiD49 and MiD51 have a different expression pattern by tissue type and age. In most adult cells, MiD49 expression is higher than MiD51 expression, except for skeletal and heart muscle cells. Furthermore, MiD51 is highly expressed in fetal tissue [51]. In tissue with high MiD51 expression, it is thought to have a regulatory effect on mitochondrial fission dynamics by increasing fission when cytoplasmic ADP levels are high during metabolic stress. Given that MiD49 and MiD51 are localized at mitochondria-associated endoplasmic reticulum sites (MAMs), they play a supportive but not essential role in ER being wrapped around the mitochondria during fission initiation, which reduces mitochondrial diameter to enable Drp1 helix formation [47,48]. Interestingly, overexpression of MiD49 and MiD51 causes Drp1 sequestration, resulting in reduced fission activity and increased number of fused mitochondria in vitro [52]. In addition to mitochondrial fission, Drp1 adaptors are involved in cell cycle regulation, with MiD49 and MiD51 being important in apoptotic Drp1 recruitment to the mitochondrial membrane along with Bax, and interestingly with Mfn2, resulting in cytochrome c release and concurrent mitochondrial fission [29,53]. Furthermore, loss of MiD49 and MiD51 were shown to increase resistance to apoptotic stimuli [54]. For more details of mitochondrial fission and fusion in OC, see Section 3.3, with preclinical and human data for OC found in Section 3.3 (Figure 3).

### 2.5. Mitochondrial Fusion

Mitochondrial fusion, opposing fission, joins two mitochondria together, resulting in increased OXPHOS capacity. The key enzymes included in fusion are mitofusin 1 and 2 (Mfn1, Mfn2), and optic atrophy 1 protein (OPA1). While Mfn1 and Mfn2 are involved in MOM fusion (via homotypic, Mfn1-Mfn1/Mfn2-Mfn2; or heterotypic, Mfn1-Mfn2 connections), OPA1 fuse the internal mitochondrial membranes (MIM) connecting to other OPA1 proteins [27]. For more details of mitochondrial fission and fusion in OC, see Section 3.3, with preclinical and human data for OC found in Section 3.3 (Figure 4).

#### 2.5.1. Mfn1 and Mfn2

Although Mfn1 and Mfn2 are structurally similar GTPases, Mfn2 has a Ras-binding domain [55], and often inhibits Ras downstream signaling via sequestration [56]. Subsequently, this results in decreased cell proliferation and cell-cycle arrest, showing a tumor suppressor effect in many types of malignancies [55,57]. Regarding their mitochondrial fusion activities, Mfn2 is a less efficient than Mfn1, the latter showing about an 8–20 fold higher GTPase activity [58,59]. Heterotypic Mfn1-Mfn2 complexes are, however, more efficient than the homotypic Mfn2-Mfn2 or even Mfn1-Mfn1 complexes for mitochondrial fusion [60]. Mfn2, although less important in mitochondrial fusion, is able to promote a stronger ER–mitochondria tethering than Mfn1, which is crucial in inter-organelle Ca^2+^ transport [61]. Furthermore, both Mfn1 and Mfn2 can interact with the MIM fusion protein OPA1, helping to create a physical connection between the MOM and MIM to complete mitochondrial fusion [62].

#### 2.5.2. OPA-1

OPA1 has eight different isoforms, generated by alternative splicing of 3 (exon 4, exon 4b, exon 5b) out of the 31 exons, which are ubiquitously expressed in different amounts in different tissues [63]. When OPA1 is transferred to the MIM, the so-called mitochondrial targeting sequence is cleaved by the mitochondrial processing peptidase (MPP), creating the long, transmembrane OPA1 isoforms (L-OPA1) [63]. L-OPA1, which is the membrane-bound, fusion-active form, can undergo further proteolytic processing by OMA1 and YME1L peptidases, which cleave the S1 or S2 sites, respectively, producing the short, soluble OPA1 isoforms (S-OPA1), located in the intermembrane space. Interestingly, while the OMA-cleaved S1 site (exon 5) is present on all OPA1 isoforms, the YME1L-cleaved S2 sites (exon 5b) are only present on isoforms 4, 6, 7, and 8 [63]. Furthermore, YME1L also cleaves S3 (encoded by exon 4b), found in isoforms 3, 5, 6, and 8 [64]. For efficient MIM fusion, both L-OPA1 and S-OPA1 isoforms are necessary, with S-OPA1 strongly stimulating L-OPA1-dependent MIM fusion after the established hemifusion state. Notably, L-OPA1 alone is sufficient to make a very low level of mitochondrial fusion, whereas S-OPA1 by itself is not [65]. High S-OPA1 concentration, however, can also disrupt L-OPA1 activity, resulting in reduced fusion [65], highlighting the sensitivity of mitochondrial fusion based on the of L-OPA1 and S-OPA1 [64]. Besides its function in mitochondrial fusion, OPA1 has multiple other roles in cellular homeostasis and function: 1. It stabilizes MIM cristae and is keeping them tight, preventing cytochrome c release and subsequent apoptosis initiation without interfering with Bax and Bak [66]. 2. OPA1 mutations were shown to increase mitophagy [67]. 3. OPA1 is directly involved in the maintenance of mitochondrial DNA integrity: OPA1 silencing results in mtDNA depletion and replication inhibition [68]. 4. It affects intracellular Ca^2+^ signaling by regulating mitochondrial Ca^2+^ uptake/ ER–mitochondria transfer, and by stabilizing the mitochondrial calcium uniporter 1 (MICU1). Whether OPA-1 enhances or reduces mitochondrial Ca^2+^ uptake, it seems to be cell- and model-dependent [69,70]. 5. Cleaved OPA1-exon4b isoforms maintain mitochondrial metabolism by binding directly to the mtDNA D-loop, enhancing the transcription of key mitochondrial genes [71,72]. It accelerates tumor growth and NFκB-dependent angiogenesis by increasing angiogenetic gene expression independent from its role in improving mitochondrial oxygen delivery in in vivo and in vitro angiogenesis and breast cancer/ melanoma models [70,73] (Figure 4).

### 2.6. Mitophagy

Mitophagy is important in reducing the number of mitochondria and in removing dysfunctional mitochondria to maintain homeostasis [27]. During non-selective autophagy, for example, due to starvation, mitochondria can be randomly involved in autophagy along with other cellular organelles and cytoplasm. On the other hand, selective mitophagy specifically targets mitochondria for autophagy. There are two major selective mitophagy pathways described: 1. ubiquitin-mediated, and 2. receptor-mediated pathways. Here, we describe these briefly, but for further details you can read previous reviews, such as the one from Xia and colleagues (2025) [74]. For more details of mitophagy in OC, see Section 3.3, with preclinical and human data for OC found in Section 3.3.

#### 2.6.1. Ubiquitin-Mediated Selective Mitophagy

There are parkin-dependent and parkin-independent mitophagy pathways that are ubiquitin-mediated. In the parkin-dependent pathway, phosphatase and tensin homolog (PTEN)-induced putative kinase 1 (PINK1) accumulates in the MOM due to its inability to be transferred through the MIM to be cleaved. This increased PINK1 presence at the MOM induces parkin phosphorylation and recruitment from the cytoplasm [27]. Activated parkin subsequently ubiquitinates mitochondrial proteins, such as Mfn1, Mfn2, voltage-dependent anion channel (VDAC), and Miro1, tagging them along with the mitochondria for elimination. Once ubiquitinated, these OMM proteins bind to autophagy adaptor proteins (p62, Optn, and Ndp52), which bind the cargo to the microtubule-associated protein light chain 3 (LC3) proteins in the autophagosome membrane via their LC3-interacting regions (LIR) [75]. Besides PINK1, E2 ubiquitin-conjugating enzymes (such as UBE2D and UBE2L3) can activate parkin, with another enzyme in this group, UBE2R1, antagonizing the parkin effect, and UBE2N mediating mitochondrial clustering for mitophagy [76,77]. To make this process more complex, other E2 ubiquitin-conjugating enzymes, such as glycoprotein 78 (Gp78), mitochondrial E3 ubiquitin ligase 1 (MUL1), smad ubiquitination regulatory factor 1 (SMURF1), and ariadne RBR E3 ubiquitin protein ligase 1 (ARIH1) can tag mitochondrial proteins for degradation besides parkin [78]. Interestingly, ubiquitination of the fusion proteins Mfn1 and Mfn2 makes mitophagy more effective by fragmenting the mitochondria, making them an easier target [27,79].

#### 2.6.2. Receptor-Mediated Selective Mitophagy

The second receptor-mediated mitophagy pathway is via receptors on the MOM [activating molecule in Beclin1-regulated autophagy 1 (AMBRA1), Bcl-2 family adenovirus E1B 19 kDa-interacting protein 3 (BNIP3), BNIP3L/Nip3-like protein X or Bnip3L (NIX), Bcl-2-like protein 13 (BCL2L13), FUN14 domain containing 1 (FUNDC1)], and MIM [Prohibitin 2 (PHB2), cardiolipin], which directly bind to LC3 after activation via phosphorylation and/or dephosphorylation [80]. Interestingly, besides activation by parkin-independent phosphorylation, AMBRA1 can be recruited in a parkin-dependent manner, representing an intersection for the two mitophagy pathways [81,82].

### 2.7. Membrane-Associated Ring-CH-Type Finger 5 (MARCH5)

Further mitochondrial stress markers include the E3 ubiquitin ligase membrane-associated ring-CH-type finger 5 (MARCH5, also known as MARCH-V or Mitol), located on the MOM. It affects mitochondrial dynamics and other mitochondrial functions through various proteins: 1. MARCH5 degrades mitochondrial pyruvate carrier 1 (MPC1), which reduces pyruvate transport to the mitochondria that fuels OXPHOS, promoting anaerobic glycolysis. 2. It removes misfolded mitochondrial proteins, such as the mutant superoxide dismutase1 (mSOD1). 3. It targets FUNDC1, especially under hypoxic conditions, to inhibit mitophagy. 4. Also, MARCH5 degrades Mfn1 via ubiquitylation, resulting in decreased fusion. 5. It also ubiquitylates Mfn2, promoting Mfn2-ER interaction for Ca^2+^ transfer. 6. Its Drp1 ubiquitination by MARCH5 results in decreased Drp1 activation without its degradation. 7. MARCH5 also ubiquitinates and induces degradation of the fission protein MiD49, inhibiting stress-induced mitochondrial fission when apoptosis inducers and mitochondrial toxins were used to induce stress [83,84,85,86,87,88]. To make MARCH5 effects on mitochondrial dynamics more complex, MiD49 ubiquitination by MARCH5 is negatively regulated by Drp1 and Mff [89], highlighting the dominance of Drp1 and Mff in the regulation of mitochondrial fission. Furthermore, mitochondrial fission is increased in MARCH KO cells due to reduced MiD49 degradation with no significant effect on mitochondrial fusion or metabolism [88]. Also, MARCH seem to have the most significant effect on MiD49 amongst mitochondrial dynamics-related proteins [88]. In general, MARCH5 has a critical role in the maintenance of mitochondrial homeostasis. For more details about MARCH5 in OC, see Section 3.3 and Section 3.4.

### 2.8. Mitochondrial Ca^2+^ Homeostasis Regulation

#### 2.8.1. Mitochondria-Associated Endoplasmic Reticulum Membranes (MAMs)

MAMs refers to close (10–50 nm wide) contact sites between the MOM and the ER, connected by tethering proteins [56,90,91]. These connections facilitate the communication between the two organelles without altering the cytoplasmic environment [92]. Their function in most cells include the regulation of lipid metabolism and transfer, Ca^2+^ homeostasis, ROS production, and autophagy [92,93]. This section briefly discusses the Ca^2+^ homeostasis, and for further review we refer to recent reviews from Csordas and co-workers [92], Liu and co-workers [94], or Doghman-Bouguerra and co-workers [95].

#### 2.8.2. Voltage-Dependent Anion Channel (VDAC) and the Mitochondrial Ca^2+^ Uniporter (MCU)

Under normal conditions, mitochondria do not store Ca^2+^. Various mitochondrial proteins are, however, stimulated by Ca^2+^, such as the ATP synthase and some Krebs cycle enzymes [92,95]. Notably, largely increased Ca^2+^ influx to the mitochondria results in mitochondrial swelling, which may result in apoptosis induction via cytochrome c release [95]. From the mitochondrial side, the MOM-located VDAC and the MIM-located MCU are the main regulators of the intramitochondrial Ca^2+^ levels. VDAC, which is an important structural protein in the mitochondria, regulates the transport of various ions and water-soluble metabolites through the MOM [96]. MCU, on the other hand, is located on the MIM, forming a complex with multiple other proteins, such as mitochondrial calcium uniporter 1 &2 (MICU1 &2), mitochondrial calcium uniporter dominant negative subunit beta (MCUb), essential MCU regulator (EMRE), and mitochondrial calcium uniporter regulator 1 (MCUR1) transporting Ca^2+^ in the mitochondrial matrix [95]. Of these proteins, MICU1 seems to be the sensor of the high Ca^2+^ concentration (i.e., ≥500 nM) in the intermembrane space, which undergoes a conformational change that results in Ca^2+^ influx to the mitochondrial matrix [97]. The low Ca^2+^ affinity of the MCU complex makes it the gatekeeper of the intramitochondrial Ca^2+^ concentration [95]. For more details on these in OC, see Section 3.3 and Section 3.4.

#### 2.8.3. Inositol 1,4,5-Trisphosphate Receptor (IP3R), Glucose-Regulated Protein 75 (GRP75), and Sigma 1 Receptor (Sig1R)

From the ER side, IP3R is the major ion channel, located on the ER membrane, connecting to VDAC on the mitochondria. While GRP75 helps to connect IP3R3 with VDAC, Sig1R along with other ER chaperons are important in stabilizing IP3R3. These therefore enhance Ca^2+^ transfer to the mitochondria. For more details on these in OC, see Section 3.3 and Section 3.4.

#### 2.8.4. Vesicle-Associated Membrane Protein-Associated Protein B (VAPB) and Protein Tyrosine Phosphatase Interacting Protein 51 (PTPIP51)

Similar to the Mfn2-Mfn2 interaction between mitochondria and ER, interaction between tethering proteins VAPB on the ER, and PTPIP51 on the MOM, facilitate Ca^2+^ trafficking from the ER to the mitochondria by bringing the two organelles closer to each other [98]. These are supported by the observation of reduced mitochondrial Ca^2+^ uptake with the loss of VAPB or PTPIP51 proteins [98]. For more details on these in OC, see Section 3.3 and Section 3.4.

### 2.9. A-Kinase Anchoring Proteins (AKAPs)

AKAPs is a diverse group of scaffolding proteins that bring together protein kinase A (PKA) and other kinases, phosphatases, and effector proteins into signaling complexes in specific intracellular localizations [99,100]. What all AKAP proteins have in common is that they have a PKA-binding domain, other protein–protein interaction domains, and a cellular localization signal that places them in specific subcellular compartments [100] (Welling, 2008). Mitochondria-associated AKAPs include AKAP1, AKAP2, acyl-CoA-binding domain-containing 3 (ACBD3; aka PAP2, GCP60), sphingosine kinase interacting protein (SKIP), Wiskott–Aldrich syndrome protein family verprolin-homologous protein 1 (WAVE-1), and Rab32 [101]. Interestingly, fusion protein OPA1 is also in the AKAP protein group. Also, for more details on these in OC, see Section 3.3, Section 3.4 and Section 3.5.

#### 2.9.1. AKAP1

AKAP1, located on the MOM, interacts with various proteins and mRNAs, including Drp1, inhibiting mitochondrial fission [102]. AKAP1 is also important in localized translation of ~20% of nuclear-encoded mitochondrial genes on the MOM [103].

#### 2.9.2. AKAP2

AKAP2 is also located on the MOM [104], and it integrates PKA and G protein signaling [105]. It has a complex role in promoting angiogenesis and in inhibiting apoptosis via upregulation of VEGF and bcl-2, respectively [106].

#### 2.9.3. ACBD3

Although ACBD3 is primarily described to be Golgi-membrane-bound, it can also be found on the mitochondria [107]. This multifunctional protein is involved in Golgi organization, ER to Golgi transport, membrane trafficking, and mitochondrial cholesterol transport and therefore in steroidogenesis [28,108].

#### 2.9.4. SKIP

SKIP is found in the mitochondrial intermembrane space and in the mitochondrial matrix (latter surrounded by the MIM), recruited in various mitochondrial protein complexes, including one (mitochondrial contact site and cristae organizing system/MICOS complex) with coiled-coil-helix-coiled-coil-helix domain containing protein 3 (ChChd3) in the MIM, where it phosphorylates ChChd3, maintaining mitochondrial crista integrity and function [109,110]. It may also be involved in mitophagy, as inhibiting its target, sphingosine kinase 1, disrupts Pink1 and Bnip3l/Nix-mediated mitophagy [111]. Also, SKIP knockout or silencing promotes apoptosis, mitochondrial fragmentation [109,112], and causes metabolic dysfunction [112,113].

#### 2.9.5. WAVE-1

WAVE-1 is an actin-bound protein under normal conditions, involved in the regulation of actin cytoskeletal dynamics and cellular motility [114,115]. It is, however, localized to the mitochondria during apoptosis induction, where it interacts with the antiapoptotic bcl-2 protein, releasing it from the mitochondria to the cytosol [116]. Furthermore, during neuronal ischemia-induced apoptosis, WAVE-1 forms a complex on the mitochondrial MOM with Bcl-xL and Pancorton-2, allowing Bax binding, mitochondrial pore formation, and cytochrome-c release, and subsequent caspase activation and cell death [117]. Interestingly, WAVE-1 also forms a complex that includes the pro-apoptotic BAD protein and glucokinase in hepatocytes, where it enhances apoptosis and alters glucose metabolism [118].

#### 2.9.6. Rab32

Rab32 is found in the MAM, where it can phosphorylate Drp1 on the Ser656 site, causing Drp1 inactivation and subsequently increased mitochondrial fusion. Drp1 Ser656 phosphorylation also leads to apoptosis resistance in vitro [119,120]. Furthermore, increased Rab32 expression decreases mitochondrial Ca^2+^ uptake from the ER [119]. Rab32 also induces autophagy of mitochondrial-proximal ER membranes in the MAM [121].

### 2.10. Mitochondrial Unfolded Protein Response (UPRmt)

#### 2.10.1. Mitochondrial Unfolded Protein Response (UPRmt)

Increased oxidative stress results in increased mitochondrial protein misfolding, triggering a protective cellular response, called UPRmt [122]. In many ways, it is similar to ER stress response; however, UPRmt is not fully understood in mammals. For changes in OC, see Section 3.4 and Section 3.5.

#### 2.10.2. Activating Transcription Factor 5 (ATF5) and ATF4

The nuclear transcription factor ATF5 is a key effector of the UPRmt, which translocates to the nucleus from the mitochondria upon mitochondrial stress, promoting cell survival [123]. ATF5 induces the expression of a large number of genes, including the antiapoptotic *BCL2* or *MCL1* genes, and in some models, some pro-apoptotic genes, such as the cyclin D1 gene *CCND1*, and genes associated with increased (*DVL1*, *EGR1*, vascular endothelial growth factor [VEGFA] and other hypoxia-inducible factor 1 [Hif1] target genes) or decreased (ID1) tumorigenesis, and increased (*ITGα2*, *ITGβ1*) tumor cell invasion [124]. Furthermore, in the canonical mammalian UPRmt pathway, the expression of OMA1, which cleaves L-OPA1, also cleaves the long DAP3 binding cell death enhancer 1 (L-DELE1) to its short form, S-DELE1 in the mitochondrial matrix. Heme-regulated inhibitor (HRI), an eIF2α kinase, is activated by these cytosolic S-DELE-1 fragments, subsequently increasing the expression of ATF4 [125,126]. ATF4 then promotes integrated stress response (ISR) by increasing the synthesis of variable metabolic pathway proteins, such as the ones in glutamine transport, amino acid, or lipid synthesis, besides others [127,128,129,130]. ISR also increases ATF5 translation. For changes in OC, see Section 3.4 and Section 3.5.

#### 2.10.3. CCAAT Enhancer Binding Protein (C/EBP) Homologous Protein (CHOP)

Another important mammalian UPPRmt protein is CHOP, which is mostly known as an ER-stress protein. Mitochondrial stress, however, also can induce increased CHOP transcription, which, besides several other genes, increases the expression of ATF5.

Further details on the different UPRmt pathways are discussed in recent reviews, such as the one from Torres and co-workers [131] or from Charmpilas and co-workers [132]. For changes in OC, see Section 3.4 and Section 3.5.

### 2.11. ‘Mitokines’

Growth differentiation factor 15 (GDF15) and fibroblast growth factor 21 (FGF21). ‘Mitokines’ are not mitochondrial proteins per se, but are cytokines that are released from cells following significant mitochondrial stress and UPRmt [133]. Two well-known ‘mitokines’ are FGF21, and the transforming growth factor-β (TGF-β) superfamily protein GDF-15. They are often increased in various tumors, in infections, in inflammatory conditions, cardiovascular disease, and obesity [133,134,135]. For changes in OC, see Section 3.4 and Section 3.5.

## 3. The Role of Mitochondrial Stress in Ovarian Cancer (OC)

In this section we review mitochondrial stress in ovarian cancer in the same order as the stress markers were discussed in the mitochondrial stress section.

Mitochondrial stress markers are emerging as potential prognostic indicators in ovarian cancer, reflecting the cancer’s metabolic adaptations and response to treatment. Ovarian cancer development is often linked to increased oxidative stress (OS) and a compromised antioxidant system. Oxidative stress can promote cancer cell proliferation, metastasis, and chemoresistance, while also inducing apoptosis in cancer cells.

Alterations in mtDNA, OXPHOS, and mitochondrial dynamics are linked to cancer progression, metastasis, and chemoresistance. Specifically, changes in mitochondrial mass, ROS production, biogenesis, and mitochondrial dynamics are being investigated as potential biomarkers that are increasingly recognized for their potential role in predicting ovarian cancer prognosis.

### 3.1. Metabolic Changes

The mitochondrial electron respiratory chain consists of four enzyme complexes (I, II, III, and IV) in which the stability of the protein subunits is essential to maintain mitochondrial function. The mitochondrial electron transport chain is often affected in carcinogenesis [136]. Excess reactive oxygen species are produced when the electron transport chain suffers damage, which is also an important factor in cellular damage [137].

Previous in vitro and in vivo studies have shown variable results on OXPHOS and anaerobic glycolysis in OC. While some studies showed that OC, especially the more invasive cell lines, are OXPHOS-dependent, other ones proved that high-grade serous OC has a significantly increased glycolysis, with an in vitro study reporting a mixture of cells that are OXPHOS-dependent, anaerobe glycolysis-dependent, or showing a mixed metabolic phenotype [138,139]. Increased OXPHOS was described in chemotherapy-resistant and cisplatin-resistant cell lines as well, with many of them also showing enhanced glycolysis. In line with this, in human studies, there were mixed results regarding the change in the metabolic activity of OC tissues: many showed increased glycolytic activity with or without increased OXPHOS, and higher grade tumors had increased OXPHOS in many cases [139]. Importantly, chemotherapy can change the metabolic profile of the tumor cells [140]. See Table 2 for more details.

Additionally, recent evidence highlighted the possible role of several OXPHOS components in EOC prognosis; however, their contribution might be different on the transcriptional and translational level: (1) mitochondrial translation-related genes; (2) high transcript levels of mitochondrial ribosomal small (MRPS) and large (MRPL) subunit proteins (MRPS12, MRPS14, MRPL15, MRPL34, and MRPL49) are suggested as novel prognostic markers predicting reduced overall survival in OC patients [141]. Additionally, a single nucleotide polymorphism (SNP) of the mitochondrial elongation factor Tu (*TUFM*) gene is associated with increased EOC risk [142]. Interestingly, the high-OXPHOS-expressing tumors have shown an increased response to conventional chemotherapy and are associated with better prognosis in HGSOC patients, likely because they have a higher cellular oxidative stress than cells with lower OXPHOS.

To evaluate the OXPHOS alterations in high-grade EOC tumor samples and OC cell lines, proteomic analysis was performed in a previous study, investigating the expression of proteins involved in mitochondrial energy metabolism, biogenesis, and ROS formation. Furthermore, altered mitochondrial biogenesis and OXPHOS proteins were investigated in tissues from HGSOC biopsies, OC cell lines, and in peritoneal ascites samples by immunoblotting and flow cytometry. OXPHOS was investigated by looking into the expression of the different subunits of ETC complexes (complexes I–IV) and the ATP synthase (complex V). The most evident changes were observed in subunits of complex II and IV in the HGS OC biopsies [143]. The heterogeneity and the increased OXPHOS subunit expression in most of the HGSOC biopsies was also reported by other authors [144]. See more details in Table 2.

#### 3.1.1. Ubiquitin–Cytochrome C Reductase Rieske Iron–Sulphur Polypeptide 1 (UQCRFS1)

UQCRFS1 is an essential subunit of mitochondrial respiratory complex III. Deleting the gene encoding UQCRFS1 causes mitochondrial complex III deficiency, cardiomyopathy, and alopecia totalis. Preliminary evidence has shown that UQCRFS1 is highly expressed in both human gastric and breast adenocarcinomas, and it has been identified as a prognostic marker for melanoma. Furthermore, UQCRFS1 expression level showed positive correlation with OC prognosis in a previous study. It affects tumor cell proliferation, cell cycle regulation, apoptosis, and DNA damage, possibly related to the AKT/mTOR signaling pathway. UQCRFS1 is therefore a possible prognostic biomarker for OC [145].

#### 3.1.2. DNAJC15

DnaJ heat shock protein family (Hsp40) member C15 (DNAJC15), also known as methylation-controlled J protein (MCJ), is a co-chaperon that belongs to the DNAJC subfamily, encoded by the *DNAJC15* gene. This protein is localized in the MIM, where it interacts with the TIMM23 translocase complex, enhancing the ATPase activity of mitochondrial heat shock protein 70 [146]. This favors the transport of proteins that lack mitochondrial targeting sequences, thus contributing to the biogenesis of these organelles. Its role is in regulating OXPHOS efficiency, oxidative stress response, and lipid metabolism. DNAJC15 can trigger ferroptosis in OC cell lines by augmenting lipid peroxidation and by lowering antioxidant defense mechanisms. Recently, it has been proposed that the loss of DNAJC15 correlates with cisplatin (CDDP)-resistance onset in OC [147].

#### 3.1.3. Mitochondrial Sirtuins

Mitochondrial respiration is a major source of ROS, and mitochondrial dysfunction can exacerbate ROS production. Mitochondrial sirtuins (SIRT3, SIRT4, and SIRT5) play pivotal roles in promoting this homeostasis by regulating numerous aspects of mitochondrial metabolism in response to environmental stressors. Mitochondrial sirtuins, such as SIRT3 and SIRT5, have been shown to be differentially expressed in OC and may be associated with prognosis [148].

#### 3.1.4. Mitochondrial Peroxiredoxins

Mitochondrial peroxiredoxins (Prdx3 and Prdx5), which are ROS scavengers, are also implicated in OC progression. An increased production of Prdx3 and Prdx5 has been shown to correlate with the development of chemotherapy-resistance in certain types of cancers. Understanding the interplay between mitochondrial Prdxs and ROS in carcinogenesis can be useful in the development of anticancer drugs with better proficiency and decreased resistance. However, more studies are required to elucidate the effect of Prdxs expression changes in OC to better understand their role as potential drug targets [149].

#### 3.1.5. Mitochondrial Pyruvate Carrier (MPC)

In OC, the loss of the MPC is a common event linked to more aggressive and therapy-resistant disease. When MPC is lost, the cell cannot properly use pyruvate for mitochondrial energy production (OXPHOS), which forces it to adapt its metabolism for survival, leading to increased reliance on other fuel sources, such as glutamine [150]. This metabolic shift turns OC cells to be ‘addicted’ to glutamine, which is associated with more prominent cancer cell proliferation, resistance to treatment, and a more aggressive phenotype. Loss of MPC also drives the production of proline, a key component of collagen synthesis, which supports cancer cell proliferation and other aggressive traits. Because of this strong link between MPC loss and cancer aggressivity, targeting metabolic pathways that are upregulated during MPC loss is a potential new therapeutic strategy. Low expression of MPC1 is observed in many OC cases, which correlates with a poor prognosis [150]. See more details in Table 2.

**Table 2 ijms-27-00342-t002:** Mitochondrial metabolic markers and mtDNA replication/transcription changes in ovarian cancer (OC), and their therapeutic and/or prognostic relevance. Abbreviations: a/w: associated with; OC: ovarian cancer; PCT; placebo-controlled trial.

Protein Name and Inhibitors	Protein Function	Alterations in Ovarian Cancer (OC)	Preclinical Studies	Clinical Studies
**pyruvate dehydrogenase (PDH)** **PDH is inhibited by pyruvate dehydrogenase kinase (PDK)** - PDK Inhibitor: dicholoroacetate (DCA)	TCA cycle enzyme	- Decreased PDH A1 subunit expression is a/w worse outcome in human OC tissue and in cell lines [151] - PDK1 promotes metastasis and angiogenesis in OC - increased in high-grade serous OC cells vs. benign cells [152]	- dichloroacetate inhibited OC tumor growth synergistically with metformin in vitro and in vivo [153] - dichloroacetate induced cell death in cisplatin-resistant OC cells [154] - PDK knock-out induced cell death in cisplatin-resistant OC cells [155]	-
**Citrate synthase (CS)**	TCA cycle enzyme	- increased in OC vs. benign ovarian tissue [156] - stage IV OC a/w lower enzymatic activity vs. lower stage tumors [156]	- CS inhibition by RNAi reduced cell proliferation and invasion in vitro [157]	-
**Aconitase (ACO)**	TCA cycle enzyme	- increased in high-grade serous OC vs. benign cells [152]	-	-
**isocitrate dehydrogenase (IDH)** - IDH1 Inhibitor: Ivosidenib (Tibsovo^®^)	TCA cycle enzyme	- IDH1 increased in high-grade serous OC and it is a/w shorter progression-free survival [152]	- Ivosidenib increases cisplatin sensitivity in OC [158]	-
**Alpha-ketoglutarate dehydrogenase (α-KGDH)**	TCA cycle enzyme	- increased in high-grade serous OC cells vs. benign cells [152]	-	-
**Succinate-CoA synthetase (SCS)**	TCA cycle enzyme	- succinate-CoA synthetase subunit SUCLG2 expression is increased in high-grade serous OC cell lines [152]	-	-
**Fumarase**	TCA cycle enzyme	- increased in high-grade serous OC cells vs. benign cells [152]	-	-
**Malate dehydrogenase**	TCA cycle enzyme	- increased in high-grade serous OC cells vs. benign cells [152]	-	-
**Pyruvate carboxylase (PC)**	Fuels TCA cycle by making pyruvate from oxaloacetate and CO2	- increased in OC cell lines [159]	- PC KO decreases cell proliferation in vitro [159] - poly(ADP-ribose) polymerase (PARP) enzyme tankyrase upregulates PC expression in OC cell line [160]	-
**mitochondrial respiratory chain complex I (NADH dehydrogenase)** - Inhibitors: metformin IACS-010759 imipridones (i.e., ONC201, ONC206, and ONC212)	ATP production via OXPHOS	- higher expression a/w higher mortality [156] - increased in OC tissue vs. normal tissue [161]	- metformin was an effective additive in advanced and chemotherapy-/cisplatin-/paclitaxel-resistant OC in preclinical studies [162,163,164,165,166,167,168,169,170] - metformin decreased carboplatin-induced ovarian damage in rats [171] - -metformin enhances the antitumor effect of PD-L1 inhibitors in a mouse model of OC [172] - IACS-010759 delayed tumor progression and improved survival in an orthoptic patient-derived OC xenograft model [173] - ONC201 inhibits tumor growth and decreases VEGF expression in vivo [174]	- adding metformin to platinum/taxane-based therapy did not improve survival in a randomized, phase II PCT [175] - in a phase II clinical trial the addition of metformin as a neoadjuvant & adjuvant therapy improved survival when compared to historical controls [176] - metformin with paclitaxel and carboplatin did not improve survival compared to paclitaxel or cisplatin alone in a small prospective pilot-trial [177] - metformin shows no benefit in survival in a systematic review and meta-analysis of cohort studies [178] - metformin is a/w better survival in some retrospective studies [148,179] - there is an ongoing (active) phase II clinical trial on the effect of ONC201 with paclitaxel for platinum-resistant OC (clinical trial ID: NCT04055649)
**mitochondrial respiratory chain complex II** **(succinate dehydrogenase)** - Inhibitor: lonidamine shikonin imipridones (i.e., ONC201, ONC206, and ONC212)	Krebs cycle enzyme and ATP production via OXPHOS	- increased in OC vs. benign ovarian tissue [152,156,161] - higher expression a/w higher mortality [156] - stage IV OC a/w lower enzymatic activity vs. lower stage tumors [156] - SDHA subunit overexpression leads to reduced cell proliferation in some OC cells with no effect on others [140] - SDHA subunit overexpressing OC cells also have improved survival and they are better in generating colonies in suspension [140]	- subunit SDHB knockdown increases OC cell proliferation and endothelial to mesenchymal transition in vitro [180] - subunit SDHB knockdown results in mitochondrial dysfunction in OC in vitro [180] - shikonin induces cell death in SDHA overexpressing OC cells in vitro [140] - ONC201 inhibits tumor growth and decreases VEGF expression in vivo [174]	- lonidamine added to cisplatin improved therapeutic efficacy in a small phase II study [181] - ongoing (active) phase II clinical trial on the effect of ONC201 with paclitaxel for platinum-resistant OC (clinical trial ID: NCT04055649)
**mitochondrial respiratory chain complex III (cytochrome c oxidoreductase)** - Inhibitor: antimycin A 2-N-heptyl-4-hydroxyquinoline N-oxide (HQNO) stigmatellin N-acetylsphingosine (C2-ceramide)	ATP production via OXPHOS	- increased in OC tissue vs. normal tissue [161] - higher expression a/w higher mortality [156]	- antimycin A + oligomycin + uncoupling agent, FCCP (Carbonyl cyanide 4-(trifluoromethoxy) phenylhydrazone) significantly decreases ATP production in OC cells [182]	-
**mitochondrial respiratory chain complex IV** **(cytochrome c oxidase)** - Inhibitor: cyanide sodium azide imipridones (i.e., ONC201, ONC206, and ONC212)	ATP production via OXPHOS	- increased in OC vs. benign tissue [156,161] - higher expression a/w higher mortality [156] - stage IV OC a/w lower enzymatic activity vs. lower stage tumors [156]	-	- ongoing (active) phase II clinical trial on the effect of ONC201 with paclitaxel for platinum-resistant OC (clinical trial ID: NCT04055649)
**mitochondrial respiratory chain complex V** **(ATP synthase)** - Inhibitors: oligomycin bedaquiline imipridones (i.e., ONC201; ONC206, and ONC212)	ATP production via OXPHOS	- increased complex V activity in OC cells in vitro [183] - higher expression a/w higher mortality [156]	- ONC201 inhibits tumor growth and decreases VEGF expression in vivo [174] - antimycin A + oligomycin + uncoupling agent, FCCP (Carbonyl cyanide 4-(trifluoromethoxy) phenylhydrazone) significantly decreases ATP production in OC cells [182] - bedaquiline decreases ATP production, cell migration, tumor growth, and tumor survival, and enhances cisplatin effect in OC cells in vitro [183]	- ongoing (active) phase II clinical trial on the effect of ONC201 with paclitaxel for platinum-resistant OC (clinical trial ID: NCT04055649)
**MPC1 & 2** (heterologous protein complex in the MIM) - Inhibitors: α-cyano-cinnamates (e.g., UK5099) thiazolidinediones (TZDs) (e.g., rosiglitazone, ciglitazone, troglitazone)	Pyruvate transport to the mitochondria	- lower expression of MPC1 and 2 a/w higher mortality and poorer prognosis [150,156]	- MPC1 KO and UK5099 treatment on OC cells show decreased ATP production and reduced cell growth, but the cells become more migratory and chemotherapy- and radiotherapy-resistant [184] - rosiglitazone enhances paclitaxel effect in vitro on OC cells [185] - Rosiglitazone enhances the effect of PARP-inhibitor olaparib in in vivo and in vitro OC models [186] - ciglitazone and troglitazone increase cell death and decrease proliferation in OC cell lines [187]	-
**TOP3A**	mtDNA replication	- increased TOP3A expression is a/w better survival in OC [188]	-	-
**TFAM**	mtDNA transcription	- low TFAM and PGC-1α expression is a/w chemotherapy-resistance in OC [189] - high TFAM expression, however, was also found to be a/w unfavorable prognosis in OC [190]	- inhibiting TFAM expression using siRNA inhibits cisplatin-induced apoptosis in OC cells vitro [191]; - on the contrary, TFAM inhibition by siRNA could also suppress OC cell proliferation in vitro [171]	-
**TFB1M/2M**	mtDNA transcription	- increased TFB2M is a/w poor survival rate in OC patients [192]	-	-
**PGC-1α** - inhibitor: SR-18292	mtDNA transcription activator	- PGC-1α/β are increased in an OC xenograft model [173] - PGC-1α/β expression is predictive of OXPHOS inhibition therapy in OC [173] - increased PGC-1α promotes tumor progression and chemotherapy resistance in OC [193,194] - in human OC tissue, high PGC-1α expression is a/w tumor differentiation [195] - in contrary, decreased PGC-1α was also found to promote OC tumor growth and progression [177]	- PGC-1α knockdown can induce cisplatin-resistant OC cell apoptosis [193] - SR-18292 - inhibiting PGC-1α expression using siRNA can also reduce cisplatin-induced apoptosis in vitro in another model [191]	-
**MOTS-c**	Mitochondrial sORFs, translated to small functional peptides	- MOT-c has a tumor suppressor effect in OC [196] - decreased Mots-c level in OC tissue and in blood of OC patients is a/w with poor outcome [196]	-	-

### 3.2. mtDNA and Mitochondrial Mass

mtDNA copy number per cell increases in OC; however, higher grade tumors have lower mtDNA copy number [197]. In line with this, mitochondrial biogenesis s and mitochondrial mass increase in OC [198], although there are conflicting reports on the increase or decrease in the individual mtDNA replication and transcription elements (see Table 2). Based on all the available data, the most likely cause of the discrepancies are the different models used. Given the metabolic and mtDNA translation/transcription heterogeneity of OC cells, individualized therapies, such as OXPHOS inhibition in patients with increased PGC-1α [173], are promising future therapies in OC. It is important to note that PGC-1α may have a dual role in OC, acting as either a tumor promoter or a tumor suppressor: 1. It can enhance tumorigenesis by boosting ATP production and OXPHOS, fatty acid synthesis. 2. But it has a tumor suppressive effect when increased OXPHOS results in increased ROS production and subsequent apoptosis induction. The use of inhibitors is listed in Table 2.

#### mtDNA as a Prognostic Factor

Increased mtDNA content is linked to poor prognosis and chemoresistance in OC.

Studies on ovarian tissue. A previous study revealed that the total amount of mtDNA is much higher in OC tissues than in controls, comparing frozen ovarian tissue samples from EOC patients to healthy controls, as mentioned previously. The change in mtDNA content was not related the patients’ age or tumor stages. However, the average mtDNA copy number in pathological low-grade tumors was over two-fold higher than that in high-grade carcinomas (*p* = 0.012). Change in mtDNA content therefore might be an important genetic event in the progression of ovarian carcinomas [197,199].

Cell-free mtDNA. Circulating cell-free mtDNA (ccf mtDNA) levels in peripheral blood and evaluation of its integrity showed that mtDNA is a possible biomarker for diagnosis and prognosis of EOC. The integrity of mtDNA, assessed by analyzing specific mtDNA fragments (such as the ratio of mtDNA79 and mtDNA230), can differentiate between healthy women and those with OC, and can also predict tumor progression and metastasis. The serum levels of ccf mtDNA79 and mtDNA230 were significantly higher, whereas the mtDNA integrity level (mtDNA230/mtDNA79) was strikingly lower in EOC patients than those in healthy women. Moreover, mtDNA179 showed positive correlation with the conventional CA-125 serum biomarker of OC progression. Furthermore, the levels of ccf mtDNA and its fragments has shown correlation with classic clinicopathological features of EOC: high levels of ccf mtDNA79 and ccf mtDNA230, as well as a low mtDNA integrity significantly correlated with lower (G1–2) and higher (G3) grading compared to healthy controls. Regarding FIGO stages, there were significant continuous increases in the levels of mtDNA79 and mtDNA230 from healthy women to patients with FIGO stage I–II OC, and to the patients with FIGO stage III and IV tumors in a previous study. This suggests a relevant association between the levels of mtDNA79 and mtDNA230 and OC tumor progression. Furthermore, given that the highest levels of mtDNA79 and mtDNA230 were measured in FIGO III/IV stages, showing a possible correlation of these markers with an increasing metastatic potential of the tumor. Also, a higher level of ccf mtDNA79 was described in lymph node-positive EOC patients compared to lymph node-negative EOC patients [200].

D-loop SNPs of mtDNA. The single nucleotide polymorphisms (SNPs) in the displacement loop (D-loop) of mitochondrial DNA (mtDNA) is found in many types of cancers; it can act as a possible prognostic marker in EOC as well [201]. Genetic polymorphisms in the D-loop are the predictive markers for age-at-onset in EOC patients, and may help identify EOC patient subgroups at high risk of early onset [202].

mtDNA in EOC therapy and chemoresistance. Development of chemoresistance to the mainstream paclitaxel–carboplatin treatment in EOC patients is an emerging problem in the daily clinical routine. mtDNA has a crucial role in the development of platinum resistance [203]. Mutations of mtDNA, especially the MTD4 m.10875T > C mis-sense mutation [204] that develops during paclitaxel–carbolpatin treatment is a main contributing factor in platinum resistance through regulation of mitochondrial metabolism: tumor tissues with these mutations exhibit a notably elevated lactate/pyruvate ratio compared to tissues without the mutation, indicating a shift from OXPHOS to glycolysis. Mutations in mtDNA can modulate tumor chemosensitivity through various other mechanisms, including DNA damage responsiveness, maintenance of ROS homeostasis, and alterations in mitochondrial dynamics. Concurrently, retrograde signals produced by mtDNA mutations and their subsequent cascades suggest communication with the cell nucleus, leading to the reorganization of the nuclear transcriptome, orchestrating the transcription of genes and signaling pathways associated with chemoresistance [205].

RAD 51. RAD51 or RAD51 recombinase, also known as RAD51A, which catalyzes the recognition of homology and strand exchange and enables timely DNA damage repair, is essential for the maintenance of genome integrity [206]. RAD51 has a multifaceted role in carcinogenesis, cancer progression, and anticancer drug resistance [206,207]. Epithelial–mesenchymal transition-associated drug resistance, hypoxia-mediated drug tolerance, and drug resistance in cancer stem cells all involve RAD51 resistance. RAD51 can also upregulate pro-metastatic gene expression [207]. RAD51 is a crucial component and functions at the core of homologous recombination, which recruits and allocates various mediators or regulators or interactors, such as BRCA2, PALB2, and TOPBP1 resistance [206,207]. RAD51 overexpression may lead to excessive and improper recombination, triggering genomic instability, which successively drives malignant transformation, contributes to tumor progression, and even induces anticancer drug tolerance [206,207,208,209].

### 3.3. Mitochondrial Fission/Fusion, and Mitophagy in OC

Mitochondrial morphology and dynamics are altered in OC, potentially impacting chemosensitivity and chemoresistance. Generally, fission and fusion processes play a crucial role in cancer development and chemotherapy resistance. Moreover, abnormalities in mitochondrial dynamics have been associated with metabolic alterations in OC progression as well as proteins responsible for chemosensitivity and chemotherapy resistance [210]. Mitochondrial fission generally increases in OC; however, increased fusion has also been described in some studies. These differences are likely due to tumor heterogeneity and the use of different OC models. Furthermore, hypoxia, glucose-deprivation, and chemotherapy resistance can alter mitochondrial dynamics. Most mitophagy-associated proteins also increase in OC. These highlight several further therapeutic targets in OC.

Hence, alterations in mitochondrial dynamics have been found to be significant factors in OC development and progression [199]. Specific proteins involved in mitochondrial dynamics, such as Drp1 (related to fission) and Opa1 (related to fusion), may serve as biomarkers for cancer progression and chemotherapy response [211]. For more details, see Table 3.

### 3.4. MARCH5 and Mitochondrial Ca^2+^ Homeostasis Changes in OC

MARCH5 was found to be upregulated in OC cells, contributing to their migration and invasion. The abnormal upregulation of MARCH5 is accompanied by significantly increased glycolysis in OC. Mechanistically, MARCH5 promotes glycolysis via ubiquitinating and degrading mitochondrial pyruvate carrier 1 (MPC1), which mediates the transport of cytosolic pyruvate into mitochondria by localizing in the MOM. In line with this, MPC1 expression is significantly decreased in OC, and its downregulation is closely correlated with unfavorable survival, as previously mentioned. Furthermore, in vitro and in vivo models revealed that MARCH5 upregulation-enhanced glycolysis played a critical role in the proliferation and metastasis of OC cells [87]. For more details on MARCH5 and mitochondrial Ca^2+^ homeostasis protein alterations and their use as possible therapeutic targets, see Table 4.

### 3.5. AKAPs, UPRmt, and ‘Mitokines’ in OC

Some mitochondrial AKAPs and UPRmt proteins, and ‘mitokines’ may also be targeted for OC therapy or could be used as prognostic markers. For details, see Table 5.

## 4. Special Aspects—The Role of Mitochondrial Stress Markers in Targeted Therapies: VEGF Inhibition and PARP Inhibition

### 4.1. VEGF Inhibition and Mitochondrial Stress Markers

VEGF inhibition in OC affects mitochondria by selecting cancer cells that are more dependent on OXPHOS, altering mitochondrial function. Anti-VEGF therapy creates a glucose-deprived environment in the tumor, which selects for a population of OC cells that become resistant to glucose starvation and rely more heavily on OXPHOS for survival [263]. Pathogenic mtDNA variants are also being investigated as potential biomarkers in response to anti-VEGF therapy [264]. The discovery that resistant clones become dependent on mitochondrial function suggests that combining anti-VEGF therapy with drugs targeting mitochondrial metabolism (e.g., glutaminase inhibitors) could be a potential strategy to overcome resistance.

### 4.2. PARP Inhibition and Mitochondrial Stress Markers

PARPis are used to treat recurrent OC patients due to greater survival benefits and minimal side effects. PARP1 is a DNA repair protein that regulates the growth and differentiation of cells by repairing single-strand break (SSB) and double-strand break (DSB) of the DNA. Currently, inhibition of PARP1 is considered the most effective personalized target therapy for the treatment of OC, especially in patients with relapsed platinum-sensitive OC [265,266]. PARPis, including olaparib, niraparib, and rucaparib, are recommended for the maintenance of care in patients with recurrent OC with a full or partial platinum-based chemotherapy response [267,268,269,270,271].

The therapeutic effect of PARP inhibition is generally believed to be attributed to impaired DNA repair; however, PARP inhibition can also induce mitochondrial stress in OC cells, primarily by increasing oxidative stress and altering mitochondrial function, which contributes to the anticancer effects of these drugs. By inhibiting PARP, cancer cells experience increased ROS production and subsequent DNA damage. This leads to changes in mitochondrial morphology with increased fission, and it can alter metabolic processes, ultimately contributing to cell death or a reduced ability to proliferate. Furthermore, NADPH oxidases 1 and 4 were significantly upregulated by PARP inhibition and were partially responsible for the induction of oxidative stress. Depletion of NOX1 and NOX4 partially rescued the growth inhibition of PARP1-deficient tumor xenografts [272].

#### 4.2.1. Olaparib

Olaparib induces mitochondrial stress in OC cells, a novel mechanism beyond its primary function as a PARP inhibitor. This stress involves increased ROS production, reduced ATP production, and a decrease in mitochondrial respiratory function. The drug also promotes mitochondrial fission through the CDK5/Drp-1 signaling pathway, which contributes to cell death and anticancer effects [273]. Furthermore, olaparib decreases the activity of the antioxidant enzyme GPx.

Additionally, olaparib has been shown to suppress the activity of mitochondrial complexes I and IV, resulting in a lower production of ATP. It also upregulates the expression of NADPH oxidases (NOX1 and NOX4) [272]. Inducing mitochondrial stress represents a novel way olaparib can exert its anticancer effects, adding to its known mechanism of inhibiting DNA repair in OC.

#### 4.2.2. Niraparib

Combination of cisplatin (Cis)- and niraparib (Nira)-induced cell death in a Twist-knockdown CisR OC cell model. Twist is a member of the core transcription factor helix-loop-helix (bHLH), known to be an epithelial–mesenchymal transition (EMT) master regulator associated with tumor recurrence and chemotherapy resistance [274,275,276]. Functionally, Twist was identified as a potential oncogenic protein in, and a significant contributor to cisplatin resistance and metastasis in OC [274,277,278,279,280,281]. The alteration of the DNA damage response (DDR) pathway is a predictive biomarker of platinum-based sensitivity in various cancers, including OC. Cis is alone, Nira alone, or a combination of Cis + Nira therapy increased cell death by suppressing DDR proteins in an in vitro OC model. Notably, the combination of Nira and Cis was considerably effective against cultures of Twist knockdown CisR OC cells, demonstrating enhanced ER stress, where the treatment led to the initiation of mitochondrial-mediated cell death. In addition, Cis alone, Nira alone, or Cis + Nira showed lower ki-67 (cell proliferative marker) expression and higher cleaved caspase-3 (apoptotic marker) by IHC. Hence, lethality of the combination PARPi therapy may provide an effective way to expand the therapeutic potential to overcome platinum-based chemotherapy resistance and PARPi cross resistance in OC [271].

## 5. Endoplasmic Reticulum Stress

ER stress is a critical cellular process characterized by the accumulation of unfolded or misfolded proteins within the ER lumen. When it exceeds ER’s protein-folding capacity, the disruption of the ER homeostasis activates the unfolded protein response (UPR), a highly conserved adaptive signaling network that aims to restore normal ER function by enhancing the protein folding machinery, increasing chaperone expression, temporarily halting protein synthesis, or, if the stress is excessive or prolonged, triggering programmed cell death pathways [282,283,284,285].

ER stress can be caused by various intrinsic and extrinsic factors, including increased protein synthesis levels, impaired ubiquitination and proteasomal degradation, disturbed redox homeostasis, an excess or limitation of nutrients, hypoxia, altered Ca^2+^ levels, or prolonged inflammation [286,287].

The UPR is mediated by three ER transmembrane sensor proteins that detect the accumulation of unfolded proteins and initiate downstream signaling cascades: protein kinase RNA-like ER kinase (PERK), activating transcription factor 6 (ATF6), and inositol-requiring enzyme 1 (IRE1) [284,288,289].

PERK phosphorylates the alpha subunit of eukaryotic initiation factor 2 (eIF2α), leading to the activation of activating transcription factor 4 (ATF4), which results in C/EBP homologous protein (CHOP) activation and promotes the translation of several ER stress response proteins [290]. When the ER stress is unresolved, CHOP can also mediate the UPR switch from protective to pro-apoptotic signaling by downregulating the antiapoptotic Bcl-2 [291].

Activated ATF6 is translocated to the Golgi apparatus, where it is cleaved by site-1 and site-2 proteases, producing a 50 kDa fragment (p50ATF6), which can function as a transcription factor [292].

IRE1 is a bifunctional kinase and endoribonuclease. Activated IRE1 can splice the mRNA of the transcription factor X-box binding protein 1 (XBP1), which then translocates to the nucleus and upregulates the expression of genes encoding ER chaperones and components of the ER-associated degradation (ERAD) machinery [293]. ERAD is a key quality-control mechanism in the cell, responsible for mediating the degradation of misfolded proteins. First, the misfolded substrates are recognized within the ER, then transported across the ER membrane into the cytosol, where they become ubiquitinated and degraded by the proteasome [283,294,295,296,297,298]. IRE1 signaling also activates the Jun N-terminal kinase (JNK) pathway, therefore modulating inflammatory and apoptotic responses.

The master regulator of UPR is BiP, an Hsp70 family member, which binds PERK, IRE1, and ATF6 under normal conditions, keeping them in an inactive state. When unfolded or misfolded proteins accumulate in the ER, these unfolded proteins compete with the sensors for BiP binding, causing their dissociation from BiP, becoming activated and capable to initiate the UPR [299].

Due to its critical functions, UPR dysfunction is implicated in a wide range of human diseases, including cancer [300,301], as several tumor-intrinsic and microenvironmental factors in ovarian cancer are capable to induce sustained ER stress [302].

OC cells have an extensive ER. Although several studies have examined the relationship between ER-derived pathways (i.e., the stress response, which increases ER capacity, and cell death originating in the organelle) and OC, a comprehensive summary of these studies is still lacking.

Cancer cells can adapt to adverse conditions in the tumor microenvironment by modulating the capacity of the ER to fold proteins. Increasing the activity of UPR may govern tumor adaptation to hypoxia, nutrient deficiency, and other hostile conditions, as well as may contribute to resistance against cellular stress caused by tumor therapy. Consequently, prolonged, adaptive activation of ER stress may impact cancer progression by promoting cellular survival, invasion, angiogenesis, or therapy resistance. On the other hand, excessive and prolonged ER stress may cause cell death, making it a new potential new therapeutic target [289].

### 5.1. Biomarkers Related to ER Stress in EOC

ER stress-related proteins and genes play a significant role in the biology of EOC, and they can also serve as prognostic biomarkers.

One of the most significant ER stress chaperones is GRP78, the overexpression of which promotes tumor cell growth, invasion, and resistance to chemotherapy. GRP78 levels are significantly higher in EOC tissues than in normal tissues, and high GRP78 expression is associated with poorer survival rates. Similarly, PDI overexpression indicates a poor prognosis. Both proteins have been described as important prognostic factors for EOC, so by using them in combination, we can estimate the clinical progression and aggressiveness of EOC [303,304]. Indeed, a low combined expression of the two markers above correlated with better survival rates.

Some studies also found elevated PERK and ATF6 levels in EOC, but their correlation with the survival rate is unclear and their potential role as prognostic factors needs further clarification [302].

Zhang et al. developed and validated a new prognostic risk classifier based on ten differentially expressed ER stress-related genes that predicts survival of EOC patients. The risk classifier effectively sorts patients into lower and higher risk groups, and patients in the higher-risk group had worse survival rates than those in the lower-risk group. They also found that TRPM2, an important participant in the prognostic classifier and a non-selective Ca^2+^ permeable cation channel that was upregulated in OC cells and correlated with invasion and migration, could be a promising therapeutic target [305].

### 5.2. ER-Stress-Induced Cellular Death in EOC

ER stress is able to induce several types of cell death including apoptosis, autophagy, immunogen cell death, and paraptosis. Prolonged or severe ER stress activates pro-apoptotic factors, leading to tumor cell death in ovarium cancer. This is due to the remarkable alterations of cellular homeostasis affecting Ca^2+^ stores, oxygen supply, pH changes, and oxidative stress in the tumor cells and within their microenvironment. The PERK-ATF4-CHOP pathway is responsible for apoptosis induction, and the failure of this pathway is supposed to be responsible for the cisplatin resistance of EOC. Genetic modification of the CHOP function was indeed shown to have a role in cellular survival, making the molecule a potential therapeutic target in EOC.

MicroRNA-30c-2-3p has been shown to induce apoptosis in OC cells by modulating the expression of XBP1, CHOP, and ATF4 in ER-stressed ovarian cancer cell lines. Blocking XBP1 expression was accompanied by a slight increase in CHOP and ATF4, as well as the accumulation of unfolded/misfolded proteins. Apoptosis was induced through the activation of caspases 12 and 3, which suggests that they play a crucial role in ER-stress-mediated apoptosis in EOC. This also sheds light on the potential therapeutic role of microRNA-30c-2-3p.

Inhibiting stearoyl-CoA desaturase 1 (SCD1), an enzyme involved in the synthesis of monounsaturated fatty acids, was shown to decrease the proliferation of ovarian cancer cells and induce apoptosis, while non-cancerous cells remained unaffected. SCD1 inhibition caused ER stress and subsequent apoptosis in EOC cells, highlighting the importance of lipid desaturation in tumors and establishing it as a novel therapeutic target for EOC. Increased apoptosis in ovarian cancer cells was also associated with the sustained activation of the JNK pathway, which is closely related to the IRE1 branch of UPR—therefore, prolonged activation of ER stress may lead to JNK-mediated apoptosis through the IRE1 pathway as well [61].

ER stress not only induces apoptosis when cell survival is no longer possible, but in the initial phases it stimulates autophagy, a protective cellular process that degrades damaged compartments and helps with survival. Initially, low levels of ER stress promote autophagy and inhibit tumor growth. However, in the later stages, autophagy, which is activated by prolonged cellular stress, may help tumor cells to survive and act as a counterpart to apoptosis. Bcl-2 expression, one of the regulators of autophagy, is closely related to EOC cells, and platinum resistance is positively correlated with autophagy of those. The relationship between ER stress, Bcl-2 proteins, and autophagy can influence the fate and survival of ovarian cancer cells. Therefore, it may be an important therapeutic target for EOC [306].

Ferroptosis, a form of iron-dependent programmed cell death, is also strongly related to ER stress in terms of both initiation and regulation. As these two processes can synergistically increase the death of ovarian cancer cells, they may also be a promising therapeutic target. Recently, it has been suggested that ferroptosis and ER stress are linked through the production of ROS; however, the specific mechanisms of their interaction in ovarian cancer have yet to be described [307].

### 5.3. ER Stress and Therapy Resistance in EOC Cells

Higher expression of some ER stress proteins correlates with worst patient survival (see above); therefore, they can be used as therapeutic targets in EOC [302]. GRP78 levels could affect sensitivity to cisplatin therapy in EOC by regulating autophagy and apoptosis. Modulation of ER stress and stress-related cell death is indeed a promising therapeutic approach in OC cells. The pharmacological induction of ER stress using a synthetic chalcone was also able to induce apoptosis via ROS-mediated stimulation of the UPR in both cisplatin-sensitive and -resistant OC cells, suggesting that ROS-mediated UPR induction could overcome drug resistance [308]. Hyptolide, a natural compound, was also shown to inhibit the viability of ovarian cancer cells, even in chemotherapy-resistant cases. The effect of hyptolide was mediated by the activation of GRP78 and ATF6 in response to ER stress [308]. It was also proven that combined cisplatin–hyptolide treatment increased apoptosis synergistically.

Pharmacological inhibition of the IRE1α/XBP1s signaling pathway, either alone or in combination with immune checkpoint inhibitors, is a promising therapeutic approach for cancers characterized by frequent CARM1 overexpression, including OC. Furthermore, the pharmacological targeting of the IRE1α/XBP1 pathway, particularly in conjunction with the inhibition of histone deacetylase 6 (HDAC6), is a much-needed strategy for OCs bearing mutations in the AT-rich interactive domain 1A (*ARID1A*) gene [302].

Despite the valuable approach of modulating ER stress and apoptosis in OC treatment, clinical studies are still lacking. Many factors limit the use of such compounds, like high toxicity, low bioavailability, and numerous side effects [309]. Given that pharmacologically modulating ER stress and the UPR is an effective strategy to sensitize cancer cells to conventional chemotherapy, there is a critical need to discover more selective agents with improved stability and reduced toxicity.

### 5.4. ER-Stress-Related Epithelial–Mesenchymal Transition in EOC

ER stress is able to control the EMT through numerous mechanisms, including oxidative stress, regulation of transcription factors, autophagy, and ferroptosis. In turn, EMT plays crucial role in the invasiveness of the tumor, in metastasis formation and progression, increasing aggressivity of cancer cells. The ER-stress-related induction of EMT was also suspected in ovarian cancer. A tumor suppressor candidate 3 (TUSC3), an ER localized protein responsible for N-glycosylation of proteins, is often lost in epithelial cancers, thus triggering ER stress and inducing hallmarks of EMT in EOC cells [310].

### 5.5. ER Stress and Tumor Microenvironment, Immune Responses

ER stress may be responsible for altering the tumor microenvironment in EOC cells, as it induces angiogenesis, suppresses immune surveillance, and promotes tumor progression. This is ensured by the interaction of the UPR with other signaling pathways, such as the mTOR, NF-κB, and MAPK pathways. For example, the IRE1α-XBP1 pathway has been shown to play a critical role in immune evasion by activating XBP1 in infiltrating dendritic cells within the tumor microenvironment. This alters their antigen-presenting ability, leading to impaired antitumor T cell responses [311] and reduced T cell infiltration [312], which further decreases antitumor immunity.

### 5.6. The Role of ER Stress in Ovarian Cancer Treatment

ERS and UPR play key regulatory roles in ovarian carcinogenesis and progression and may be important therapeutic targets for ovarian tissue and cancer-like diseases. Various clinically used and novel pharmacological agents are reviewed regarding their effect on ERS in OC progression and their role in treatment [302,313].

Regarding currently available therapies, ER stress and UPR have a crucial role in platinum resistance and PARP inhibition. GRP78, the most abundant and well-characterized glucose-regulated protein, is a major stress-inducible chaperone localized to the ER. Cisplatin-sensitive cells with low expression of GRP78 tended to undergo senescence easily when compared with cisplatin-resistant OC cells following a dose-gradient cisplatin exposure. Intervention against GRP78 therefore may reduce cisplatin resistance in OC [314]. Furthermore, as previously mentioned, the combination of the PARP inhibitor niraparib, cisplatin, and Twist knockdown in cisplatin-resistant OC cells could induce cell death via an ER-stress-mediated mitochondrial apoptosis pathway [271].

## 6. Conclusions

Targeting the mitochondria and the endoplasmic reticulum along with their associated pathways could offer new therapeutic strategies in ovarian cancer. Furthermore, monitoring mitochondrial and ER stress markers in ovarian cancer could provide valuable insights into disease progression, chemosensitivity, and overall prognosis, leading to improved treatment strategies and patient outcomes.

## Figures and Tables

**Figure 1 ijms-27-00342-f001:**
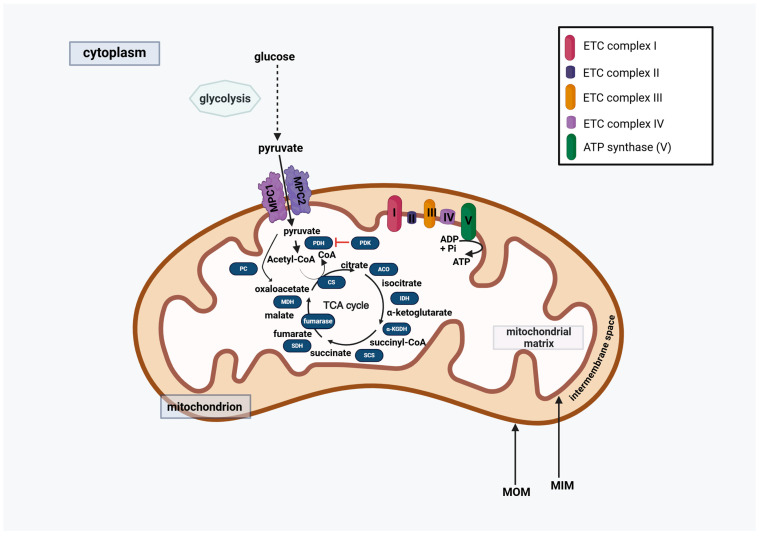
Mitochondrial electron transport chain (ETC) and the TCA cycle. Oxidative phosphorylation occurs in the mitochondria, where the mitochondrial inner membrane (MIM) ETC proteins (I–IV) transport electrons, using NADH and FADH2 to pump H^+^ into the intermembrane space to create a gradient. H^+^ ions then flow back to the mitochondrial matrix space via the ATP synthase (also known as complex V), catalyzing ATP synthesis from ADP + inorganic phosphate (Pi). The TCA cycle generates the electron carriers, NADH and FAH2, for electron donation to the ETC. The mitochondrial pyruvate carriers 1 and 2 (MPC1 and 2) are heterodimers that facilitate pyruvate transport through the mitochondrial membranes to the mitochondrial matrix, fueling the TCA cycle. Created in BioRender. Wappler-Guzzetta, E.A. (2025) https://app.biorender.com/illustrations/69307d2fcf54afbf9e5c7e82?slideId=52d01672-1809-4ee7-ab39-ed5c87ba0814 (accessed on 11 October 2025). Abbreviations: α-KGDH: α-ketoglutarate dehydrogenase; ACO: aconitase; CS: citrate synthase; IDH: isocitrate dehydrogenase; MDH: malate dehydrogenase; PDH: pyruvate dehydrogenase; PC: pyruvate carboxylase; PDK: pyruvate dehydrogenase kinase; SCS: succinate-CoA synthetase; SDH: succinate dehydrogenase. Enzymes in the TCA cycle are shown with a blue background; ├ (red) indicates blocking.

**Figure 2 ijms-27-00342-f002:**
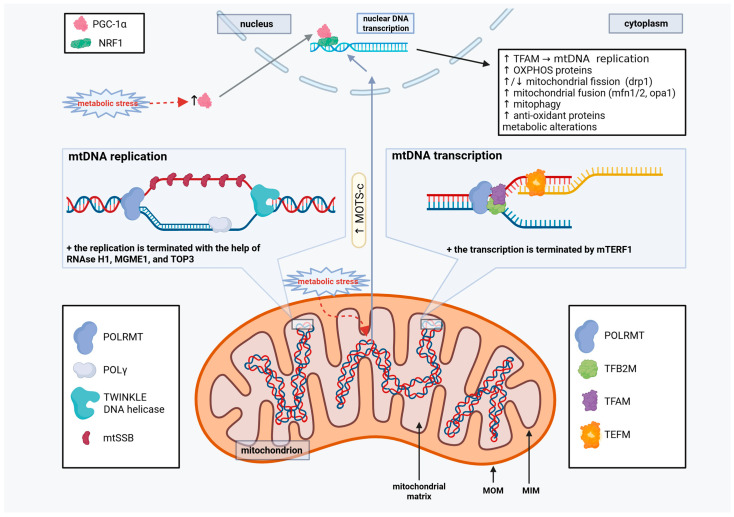
mtDNA replication and transcription. Mitochondrial DNA (mtDNA) replication is initiated by the polymerase (RNA) mitochondrial (POLRMT), while the DNA polymerase γ (POLγ) continues the DNA synthesis with the TWINKLE DNA helicase unwinding the dsDNA. The DNA strand that does not undergo the replication at the time is stabilized by mitochondrial single-stranded DNA-binding proteins (mtSBBs). When the replication is finished on both DNA strands, the RNAse H1 and mitochondrial genome maintenance exonuclease 1 (MGME1), and the topoisomerase 3A (TOP3A) help to terminate mtDNA replication. mtDNA transcription also includes POLRMT in its initiation complex, where it forms a complex with mitochondrial transcription factor A (TFAM) and mitochondrial transcription factor 2M (TFB2M). Transcription elongation factor (TEFM) is necessary for the elongation phase of the transcription, and the mitochondrial transcription termination factor 1 (mTERF1) for the termination process. Furthermore, following stress, such as metabolic stress, peroxisome proliferator-activated receptor gamma coactivator-1 (PGC-1α) can bind to nuclear transcription activators (such as NRF1) to enhance the transcription genes involved in mitochondrial biogenesis, mitochondrial fission (in some cases it can decrease drp1) and fusion, mitophagy, and antioxidant proteins, and it also alters the expression of some metabolic enzymes. Metabolic stress also increases the expression of mitochondrial-derived peptides (MDPs), such as MOTS-c, which can enhance the expression of various genes, including *TFAM* and *NRF1*. Created in BioRender. Wappler-Guzzetta, E.A. (2025) https://app.biorender.com/illustrations/69307d860f668eb5c36f2e6b?slideId=334d110a-e5b7-43cd-be3d-5e2b316d5df3 (accessed on 11 October 2025). ↑: increase; ↓: decrease; → (grey): translocation to the nucleus; → (black): results of nuclear DNA transcription and explanatory arrows.

**Figure 3 ijms-27-00342-f003:**
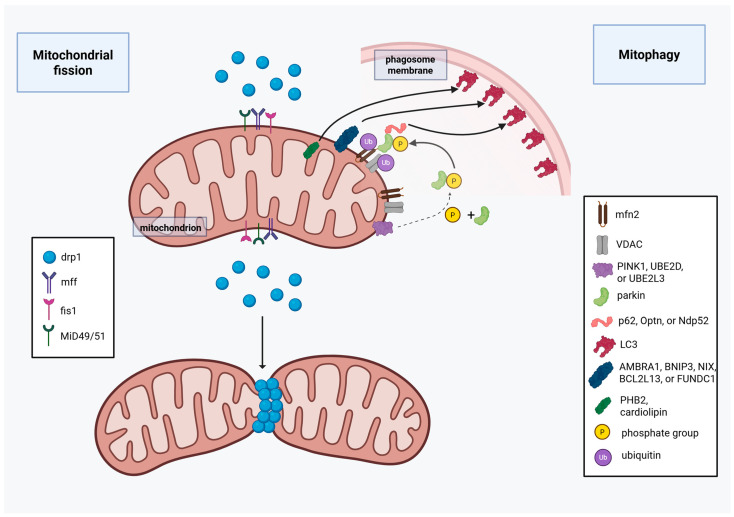
Mitochondrial fission and mitophagy—mitochondrial fission. Mitochondrial fission is largely mediated by dynamin-related protein 1 (Drp1), which is a cytoplasmic GTPase protein that translocates to the mitochondria at the time of fission, binding to the mitochondrial fission adaptors mitochondrial fission factor (Mff), mitochondrial fission 1 protein (Fis1), or the mitochondrial dynamics proteins of 49 kDa/51 kDa (MiD49/MiD51). Drp1 then forms a helical ring around the mitochondria to constrict it for fission. Mitochondrial fission adaptors can be present on the mitochondrial outer membrane (MOM) by different stimuli. Mff, for example, is active under physiological and pathological conditions, whereas Fis1 is important in stress-induced fission. Furthermore, MiD51 can bind ADP/GDP, which facilitates MiD51-associated fission, making it a good sensor of metabolic stress. MiD49, on the other hand, cannot bind ADP/GDP, and most adult tissue has a higher MiD49 expression than MiD51, except for skeletal and heart muscle cells. Also, both MiD49 and MiD51 can be found at the mitochondria-associated endoplasmic reticulum sites (MAMs), where the ER likely helps mitochondrial fission. MiD49 and MiD51 can also bind the pro-apoptotic bax, and interestingly, the fusion protein mitofusin2 (mfn2). Mitophagy can be ubiquitin-mediated and parkin-dependent or parkin-independent, and it also can be receptor-mediated. During parkin-dependent ubiquitin-mediated mitophagy, phosphatase and tensin homolog (PTEN)-induced putative kinase 1 (PINK1) accumulates in the MOM when it cannot be transferred through the MOM to be cleaved, inducing the phosphorylation and recruitment of parkin. When the phosphorylated parkin translocates to the MOM, it ubiquitinates and therefore tags mitochondrial proteins [i.e., mitofusin 1/2 (mfn1/2), voltage-dependent anion channel (VDAC)] for elimination. These ubiquitinated proteins need to then bind to autophagy adaptor proteins (p62, Optn, and Ndp52), which connect them to the phagosomal light chain 3 (LC3) protein to complete mitophagy. In addition to PINK1, E2 ubiquitin-conjugating enzymes UBE2D and UBE2L3 can also activate parkin. In the ubiquitin-mediated and parkin-independent mitophagy, other E2 ubiquitin-conjugating enzymes, such as glycoprotein 78 (Gp78), mitochondrial E3 ubiquitin ligase 1 (MUL1), smad ubiquitination regulatory factor 1 (SMURF1), and ariadne RBR E3 ubiquitin protein ligase 1 (ARIH1) ubiquitinate mitochondrial proteins other than parkin to tag them for degradation (ubiquitin-mediated and parkin-independent mitophagy is not shown on this figure, for more information see Section 2.6). In receptor-mediated mitophagy, activated proteins in the MOM [activating molecule in Beclin1-regulated autophagy 1 (AMBRA1), Bcl-2 family adenovirus E1B 19 kDa-interacting protein 3 (BNIP3), BNIP3L/Nip3-like protein X or Bnip3L (NIX), Bcl-2-like protein 13 (BCL2L13), FUN14 domain containing 1 (FUNDC1)] or MIM [Prohibitin 2 (PHB2), cardiolipin] can directly bind to the phagosomal LC3 protein. Created in BioRender. Wappler-Guzzetta, E.A. (2025) https://app.biorender.com/illustrations/69307d8f1222a70fb0157ee1?slideId=f2f208cd-3ec1-43f1-b541-5dae63abe7b7 (accessed on 11 October 2025). → (solid): translocation to the phagosome membrane/to the mitochondrial membrane or it shows the progression in mitochondrial fission. → (dashed line): shows phosphorylation.

**Figure 4 ijms-27-00342-f004:**
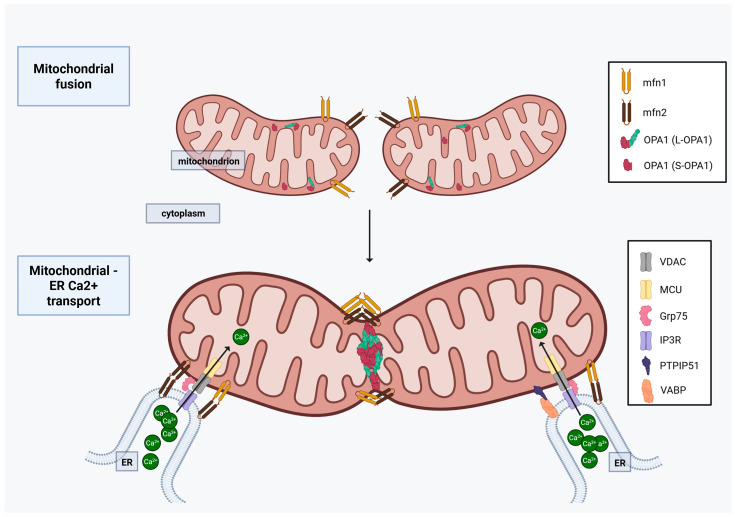
Mitochondrial fusion and mitochondrial–ER Ca^2+^ transport—mitochondrial fusion. Fusion of the mitochondrial outer membrane (MOM) is mediated by mitofusin1 and 2 (mfn1 and 2), and they can form heterotypic-(mfn1-mfn2) or homotypic (mfn1-mfn1, or mfn2-mfn2) complexes, of which the heterotypic complexes seem to be the strongest. While mfn1 is a more efficient fusion protein, mfn2 is also expressed on the endoplasmic reticulum (ER), where is helps with mitochondrial–ER tethering. The fusion of the mitochondrial inner membranes (MIMs) are mediated by OPA1, which has a long (L-OPA1) membrane-bound form and a short (S-OPA1) non-membrane bound form, both of which are important in MIM fusion. In addition to mitochondrial fusion, OPA1 plays important roles in MIM stabilization (and cytochrome c retention in the intermembrane space), increasing mitophagy, maintaining mitochondrial DNA (mtDNA) integrity, stabilizing the mitochondrial calcium uniporter 1 (MICU1), and in maintaining mtDNA transcription (not shown in the image, see text for more details). Mitochondrial–ER Ca^2+^ transport. Mitochondrial Ca^2+^ level is important in the regulation of mitochondrial metabolic activity, ROS production, autophagy, and apoptosis. Mitochondrial Ca^2+^ is mostly from the ER. The communication between the two organelles is via the voltage-dependent anion channel (VDAC) and the mitochondrial Ca^2+^ uniporter (MCU) on the mitochondrial side, and inositol 1,4,5-trisphosphate receptor (IP3R) on the ER side, along with the connector protein glucose-regulated protein 75 (GRP75). Furthermore, mfn2 (ER)-mfn1(mitochondria) and mfn2 (ER)-mfn2 (mitochondria) connections, and the interaction between vesicle-associated membrane protein-associated protein B (VAPB) (ER) and protein tyrosine phosphatase interacting protein 51 (PTPIP51) (mitochondria) help to maintain the closeness of the two organelles. Created in BioRender. Wappler-Guzzetta, E.A. (2025) https://app.biorender.com/illustrations/69307d99b18c4d13c6ad264e?slideId=f6fc3006-557d-4c29-a830-b9d41eb5bb63 (accessed on on 11 October 2025). → (solid): Ca^2+^ flux (also showing the progression in mitochondrial fusion).

**Table 1 ijms-27-00342-t001:** Pathology and molecular biology of EOC subtypes (modified table from ESMO Clinical Practice Guidelines) [5]. * Positive staining possibly in EC of uterine origin [6], although IHC is currently not used to differentiate metastatic vs. primary ECs due to lack of evidence from a larger cohort. ^$^ Napsin A is positive between 3 and 7.7% of purely EC tumors [7,8]. ^#^ Aberrant in ~7–24% of CCCs [9,10]. ^ Aberrant in ~53% of MC [11]. ^@^ Variable CDX2 expression in primary MC of the ovary (~40%) vs. metastatic GI MCs (74–100%) [12,13]. Abbreviations: CCC: clear cell carcinoma; CDX2: homeobox protein CDX2; dMMR: mismatch repair deficiency; EC: endometrioid carcinoma; EOC: epithelial ovarian cancer; ER: estrogen receptor in this table (ER refers to endoplasmic reticulum for the general article); HGSC: high-grade serous carcinoma; HNF1β: hepatocyte nuclear factor 1β; HRD: homologous recombination deficiency; IHC: immunohistochemistry; LGSC: low-grade serous carcinoma; MC: mucinous carcinoma; MSI: microsatellite instability; PAX8: paired box gene 8; WT-1: Wilms tumor.

		LGSC	HGSC	EC	CCC	MC
**IHC staining patterns**	p53 p16 WT-1 ER PAX8 Vimentin HNF1β Napsin A CDX2	Normal + + +	Aberrant + + +/− +	Aberrant/normal − − + +/− * +/− ^$^	Aberrant/normal ^#^ − − − + +	Aberrant ^/Normal − − − +/− ^@^
**Molecular alterations (decreasing prevalence from top to bottom)**		KRAS BRAF RAF	TP53 BRCA1/2 HRD	CTNNB1 ARID1A PTEN KRAS TP53 (high grade EC) MSI/dMMR	ARID1A PI3KCA PTEN MSI/dMMR	CDKN2A KRAS HER2

**Table 3 ijms-27-00342-t003:** Mitochondrial fission/fusion and autophagy protein changes in ovarian cancer (OC) and their therapeutic and/or prognostic relevance. Abbreviations: a/w: associated with; CPT1A: carnitine palmitoyltransferase 1A; EMT: epithelial to mesenchymal transition; LSD1: lysine-specific demethylase 1; MAM: mitochondria-associated membrane; MOSE: murine ovarian surface epithelial cells; OC: ovarian cancer; PARP: poly (ADP-ribose) polymerase; PART1: prostate androgen-regulated transcript 1; SREBP1: sterol regulatory element binding protein 1; TAM: tumor-associated macrophages.

Protein Name and Inhibitors	Protein Function	Alterations in Ovarian Cancer (OC)	Preclinical Studies
**Drp1** - inhibitor: Mdivi-1	mitochondrial fission	- Drp1 expression is generally increased in OC [138,212] - increased Drp1 is a/w poor prognosis and chemotherapy resistance in OC [213,214] - high expression of Drp1 alternative splice variant lacking exon 16 is associated with poor outcome in human OC tissue and cells [215] - cisplatin induces excessive mitochondrial fission and apoptosis in chemotherapy sensitive cells [216] - in contrary, cisplatin decreased Drp1 in chemotherapy-sensitive cells in vitro between day 2 and 6 in another study [217] - glucose deprivation decreases Drp1, causing fusion in OC cells in vitro [214] - OC cells use Ets1 transcription factor to increase Drp1 to promote invasion and EMT [218] - all fission and fusion proteins, including Drp11, were decreased during malignant transformation with or without hypoxia in a MOSE model of OC cells and aggregates, with fission:fusion (drp1:mfn1) ratio increasing with malignancy in adherent cells; and with the malignant cell showing increased mitochondrial mass [219]	- mdivi-1 and Drp1 silencing increases cisplatin sensitivity in OC cells under hypoxia [214] - mdivi-1 enhances cisplatin effect in platinum-resistant OC cells from peritoneal fluids and in cell lines [166,220] - mdivi-1 and drp1 silencing decreases EMT in vivo and in vitro OC models [218] - in contrary, drp1 knockdown promoted cisplatin resistance in another in vitro study [217]
**Fis1**	Drp1 adaptor protein	- all fission and fusion proteins, including Fis1, were decreased during malignant transformation with or without hypoxia in a MOSE model of OC cells and aggregates, with the malignant cell showing increased mitochondrial mass [219]	-
**Mff**	Drp1 adaptor protein	- increased in OC cell lines, promoted by CPT1A-mediated succinylation [221,222] - high mff expression is a/w poor prognosis based on human OC tissue analysis [221]	- Mff knockdown decreases MAM formation in OC cells [221] - CPT1A blocker glyburide decreases MAM formation, and reduces OC tumor cell stemness, and increases chemotherapy sensitivity [221]
**MiD49 and MiD5**	Drp1 adaptor proteins	- increased Mid49 expression in OC tissue vs. normal tissue [223]	- Mid49 knockdown inhibited cell growth, and induced apoptosis in vitro and in vivo [223]
**Mfn1**	mitochondrial fusion (MOM)	- cisplatin increases mfn1 in chemotherapy-sensitive cells in vitro between day 2 and 6 in another study [217] - all fission and fusion proteins, including mfn1, were decreased during malignant transformation with or without hypoxia in a MOSE model of OC cells and aggregates, with fission:fusion (drp1:mfn1) ratio increasing with malignancy; with the malignant cell showing increased mitochondrial mass [219]	-
**Mfn2**	mitochondrial fusion (MOM) and ER–mitochondria tethering	- cisplatin increases mfn2 in chemotherapy-sensitive cells in vitro between day 2 and 6 in another study [217] - glucose deprivation increases Mfn2 and causes fusion in OC cells in vitro [214]	- overexpression of mfn2 promotes cisplatin resistance in vitro [217]
**OPA1**	mitochondrial fusion (MIM)	- OPA1 is increased in OC tissue [198] - cisplatin increases OPA-1 in chemotherapy-sensitive cells in vitro between day 2 and 6 in another study [217] - glucose deprivation increases OPA1 and causes fusion in OC cells in vitro [214] - all fission and fusion proteins, including OPA1, were decreased during malignant transformation with or without hypoxia in a MOSE model of OC cells and aggregates, and with the malignant cell showing increased mitochondrial mass [219]	-
**OMA1**	cleaves L-OPA1 (to S-OPA1)	- OMA1 activation increases cisplatin sensitivity in vitro and in vivo [224] - p53 is required in OC cells to increase Oma1 and clear L-OPA1, causing fission and triggering apoptosis after cisplatin treatment [225]	- OMA1 silencing decreases cisplatin-induced apoptosis in an OC cell line [224] - in contrast, OMA1 knockdown inhibited L-OPA1 cleavage, increased fission, and induced apoptosis in chemotherapy sensitive OC cells in another study [202]
**YME1L**	cleaves L-OPA1 (to S-OPA1)	- upregulated in human OC, and high expression is a/w poor outcome [226]	-
**PINK1** - Inhibitor: chaetomugilin J (mycotoxin), sorafenib (VEGF and RAF inhibititor), and regorafenib (VEGFR, PDGFR, FGFR, KIT, and RAF inhibitor) [227] are multi-target kinases with PINK1-parkin inhibitory effect	parkin-dependent mitophagy protein	- increased autophagy is a/w cisplatin-resistance in OC [228] - increased levels are a/w poor prognosis [229] - PINK1- parkin-mediated mitophagy, and Drp1 expression can be induced by SREBP1, enhancing tumor cell proliferation and migration in OC cell lines [230]	- chaetomugilin J + cisplatin inhibits PINK1- parkin- mediated mitophagy and enhances apoptosis in vitro [231] - sorafenib and regorafenib induce mitochondrial damage and subsequent PINK1-parkin mitophagy in hepatocellular carcinoma cells [227], possibly contributing to the therapeutic effects of these drugs in OC, with sorafenib also targeting the mitochondrial electron transport chain complexes and the ATP synthase [232]
**p62** - Inhibitor: 3-methyladenine	adaptor protein in parkin-dependent mitophagy	- high p62 expression is a/w poor OC prognosis or higher-grade tumors in two studies [233,234]; however, other studies showed longer survival with high p62 expression [214] - OC tissue from metastatic or recurrent tumor tissues have increased p62 expression vs. patient-matched primary tumors [235] - cisplatin-resistant OC cell line has a higher p62 expression than the cisplatin-sensitive cells [236]	- mTOR inhibition with everolimus decreases p62 and enhances platinum sensitivity in high-grade serous OC in organoid models [237] - p63 knockdown or 3-methyladenine sensitized chemotherapy-resistant OC cells to cisplatin [236] - LSD1 inhibition can be sensitized by p62 knockdown in in vitro [238]
**UBE2N**	helps to direct the ubiquitinated mitochondria to the autophagosomes for degradation	- UBE2N overexpression increases paclitaxel sensitivity in OC in vitro and in vivo [239]	-
**SMURF1**	parkin-independent mitophagy	- higher SMURF1 expression is a/w poor outcome in OC patients - SMURF1 expression is higher in more aggressive OC cell line vs. less aggressive cell line	- SMURF1 downregulation inhibits OC cell invasion and migration in vitro - SMURF1 deletion decreases OC invasion in vitro [240]
**AMBRA1**	Receptor-mediated mitophagy	-	- AMBRA1 knockdown increases cisplatin sensitivity in vitro [241]
**BNIP3**	Receptor-mediated mitophagy	- high BNP3 expression is a/w poor OC prognosis [242], with another study showing the opposite, using OC tumor tissue immunohistochemical staining [243] - cisplatin treatment increases BNIP3 expression in some of the OC cell lines (A2780 and OVCAR4) [242]	- BNIP3 knockdown improves cisplatin-induced apoptosis in A2780 and OVCAR4 OC cell lines [242]
**BCL2L13**	Receptor-mediated mitophagy	- TAM-derived exosomal miRNA (miR-589-3p) binds to BCL2L13 to accelerate OC progression in vitro [244]	-
**FUNDC1**	Receptor-mediated mitophagy	- increased FUNDC1 expression is seen in OC [245] - high FUNDC1 expression in OC is a/w with better prognosis [245]	-
**PHB2**	Receptor-mediated mitophagy	- PHB2 is increased in OC tissue [198] - PHB2 overexpression decreases PARP inhibitor resistance caused by PART1 knockdown in vitro [246]	-
**LC3**	Autophagosome membrane protein	- LC3 expression is increased in high-grade serous OC vs. lower grade tumors [234] - increased LC3-II (the active form) level is a/w cisplatin resistance in human OC cell lines	- LC3 silencing sensitize OC to cisplatin in vitro and in vivo [247]

**Table 4 ijms-27-00342-t004:** Alterations in MARCH5 and proteins in mitochondrial Ca^2+^ transporter protein expressions in ovarian cancer (OC), and their therapeutic and/or prognostic relevance. Abbreviations: a/w: associated with; OC: ovarian cancer.

Protein Name	Protein Function	Alterations in Ovarian Cancer (OC)	Preclinical Studies
**MARCH5**	versatile role in mitochondrial homeostasis maintenance	- MARCH5 expression is increased in OC vs. normal tissue [248] - MARCH5 promotes OC progression by promoting aerobic glycolysis [87]	- MARCH5 silencing decreases cell migration and tumor invasion in vitro and in vivo [248]
**MCU complex proteins**	mitochondrial Ca^2+^ homeostasis regulation	- low MCUR1 expression is a/w poor prognosis in OC [249] - MICU1 overexpression is a/w poor prognosis in OC [250]	- MICU1 downregulation or silencing potentiates cell death induction in OC cells [251], and inhibits clonal growth, cell migration, and invasion in vitro [250]
**IP3R**	mitochondrial Ca^2+^ homeostasis regulation	-	- IP3R1 silencing inhibited apoptosis, whereas IP3R3 silencing induced apoptosis in OC cells in vitro [252] - TAT-fused IP3R-derived peptide sensitized OC cells to cisplatin treatment in vitro [253]
**GRP75**	mitochondrial Ca^2+^ homeostasis regulation	- GRP75 overexpression is a/w cisplatin resistance in vitro [254]	- GRP75-deficiency induces apoptosis in OC cells in vitro [254]

**Table 5 ijms-27-00342-t005:** AKAPs, UPRmt, and ‘mitokines’ in ovarian cancer (OC) and their therapeutic and prognostic relevance. Abbreviations: a/w: associated with; OC: ovarian cancer.

Protein Name	Protein Function	Alterations in Ovarian Cancer (OC)	Preclinical Studies
**AKAP2**	mitochondria-associated scaffolding protein (AKAP protein)	- AKAP2 expression is increased in OC, and it promotes cell growth and migration [255]	- AKAP2 knockdown decreased OC growth and migration in vivo [255]
**WAVE-1**	mitochondria-associated scaffolding protein (AKAP protein)	- increased WAVE1 expression is a/w poor prognosis in OC [256]	- WAVE1 silencing reduces OC cell proliferation invasion and migration in vitro [257]
**Rab32**	mitochondria-associated scaffolding protein (AKAP protein)	-	- RAB32 silencing has a tumor suppressive effect on OC cells in vitro [257]
**UPRmt**	a complex process, initiated by cellular stress	- unfolded protein response is activated in cisplatin resistance [228]	-
**ATF5**	UPRmt protein	- ATF5 expression is increased in OC tissue vs. normal tissue [258,259]	- ATF5 inhibition induces apoptosis in OC cells in vitro [258]
**CHOP**	UPRmt protein	- CHOP expression is increased in OC, with higher levels in the chemotherapy sensitive OC tissue [248]	-
**FGF21**	‘mitokine’	-	- downregulation of FGF21 increases cisplatin sensitivity in vitro [260]
**GDF15**	‘mitokine’	- GDF15 expression is high in OC tissue [261,262] and are further increased in metastatic OC tissue - GDF15 increases OC cell invasion in vitro [261]	- GDF15 knockdown or inhibition of secreted GDF15 suppressed cell invasion in vitro [261]

## Data Availability

No new data were created or analyzed in this study. Data sharing is not applicable to this article.

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
