# Peer review of "Subcellular Stress Markers in Epithelial Ovarian Cancer"

_ijms, 2025, doi:10.3390/ijms27010342_

Round 1
Reviewer 1 Report
Comments and Suggestions for Authors
This manuscript provides a comprehensive and ambitious overview of mitochondrial and ER-associated stress pathways in epithelial ovarian cancer (EOC), with particular strength in its detailed mechanistic coverage of mitochondrial dynamics, mitophagy, calcium homeostasis, and the mitochondrial unfolded protein response. The topic is timely and relevant, and the breadth of molecular detail is impressive. However, the review would benefit from stronger conceptual synthesis, clearer linkage to ovarian cancer–specific evidence, streamlined organization, and critical evaluation of biomarker and therapeutic relevance. Please see below for specific comments.
- Lines 20–21: The phrase “genetical and environmental factors, however,t metabolic components” contains typographical and grammatical errors (“genetical,” “however,t”). Please revise for clarity and professional tone.
- Lines 22–24: The abstract states that mitochondrial and ER stress are discussed as “potential biomarkers and therapeutic targets,” but no indication is given as to which markers show the strongest clinical promise. Consider adding one sentence highlighting the most compelling examples discussed later in the review.
- The abstract would benefit from a clear concluding statement emphasizing the translational relevance or future directions.
- Lines 37–41: While the histological classification is accurate, the manuscript does not sufficiently connect subtype-specific biology to later discussions of mitochondrial stress. Consider briefly flagging which subtypes (e.g., HGSC vs CCC) are known to have distinct metabolic or mitochondrial phenotypes.
- Lines 66–68: The statement “we only review non-surgical treatment options” is appropriate, but the rationale for excluding surgery should be briefly justified.
- Lines 78–87: The discussion of HRD and PARP inhibitors is accurate but largely descriptive. Consider adding a critical perspective on how mitochondrial stress or redox alterations may influence PARPi sensitivity or resistance.
- Throughout Section 2, the link to ovarian cancer is often implicit or absent. Short transition sentences indicating relevance to EOC biology would substantially improve focus.
- Lines 126–129: The description of OXPHOS versus glycolysis is accurate but generic. Please include ovarian cancer–specific data or clearly state when general cancer biology is being discussed.
- Are mtDNA copy number changes consistently observed in EOC patient samples, and are they prognostic or predictive?
- Lines 176–179: The role of PGC-1α is described broadly. Given its context-dependent role in cancer, the authors should explicitly discuss contradictory findings where PGC-1α may suppress versus promote tumor progression.
- The inclusion of mitochondrial-derived peptides is novel and interesting. However, no ovarian cancer–specific evidence is provided. Please clarify whether MOTS-c has been directly studied in EOC or whether this is a speculative extrapolation.
- Lines 264–266: Statements regarding apoptosis resistance upon loss of MiD49/MiD51 should be supported by cancer-specific data or clearly framed as general cellular observations.
- The discussion of ATF5, ATF4, and CHOP is thorough, but the manuscript would benefit from a summary paragraph synthesizing how UPRmt signaling influences tumor survival, chemoresistance, or immune evasion in EOC.
- Please clarify whether UPRmt components have been validated as biomarkers in patient cohorts or remain preclinical observations.
- Lines 622–624: The statement that higher OXPHOS correlates with better prognosis appears counterintuitive and deserves deeper discussion or mechanistic explanation.
- Lines 655–656: Claims that UQCRFS1 is a “potential therapeutic target” should be tempered unless direct targeting strategies or inhibitors exist.
- Lines 689–700: The discussion of MPC loss is strong; however, it would benefit from a brief comparison with metabolic targeting strategies currently in clinical development.
Reviewer 2 Report
Comments and Suggestions for Authors
Dear Authors,
The paper, titled "Subcellular Stress Markers in Epithelial Ovarian Cancer," provides a comprehensive overview of the most important issues related to cellular stress in ovarian cancer. The authors provide a detailed description of mitochondrial and endoplasmic reticulum stress in the pathology of ovarian cancer. The paper is well-written, yet very extensive. There are some editorial errors (Table 1 is located within the text, typos, missing spaces, and a very extensive list of references, which uses single numbers instead of number ranges, e.g., [25-28]), but any errors will certainly be pointed out by the editors. A thorough review of the content does not indicate self-plagiarism. The authors' own works are cited, but given the extensive literature, this raises no concerns. The references used represent a review of available articles spanning approximately 20 years.
However, the work lacks a materials and methods section, typical of review articles, containing PRISMA guidelines. I encourage the authors to add this, which will improve the review standards. A major strength of the work is the numerous tables and figures, which significantly facilitate and organize the manuscript's content.
Reviewer 3 Report
Comments and Suggestions for Authors
The manuscript "Subcellular stress markers in epithelial ovarian cancer" by Wappler-Guzzetta et al provides extensive review of mitochondrial and endoplasmic reticulum markers in human cells. The authors indicate defects in mitochondrial fission or fusion; TCA cycle and oxidative phosphorylation markers identified in ovarian cancer. They also include tumor suppressor and oncogenic markers mutated in different ovarian cancer types.
Mitochondrial and endoplasmic reticulum markers or mitochondria related fission/fusion mutations are common to all human cancers. What might be useful is to show ovarian specific mutations (markers) identified that stand out as a biomarker of diagnosis and therapy. Otherwise, this could be a stress response marker review of any human cancer.
Authors have less information on targeted therapies including immune checkpoint inhibitors. It will be good if a figure is included that shows recent clinical investigations of inhibitors of ovarian cancer stress markers and immune check point inhibitors with or without chemo or radiation therapies and survival data (Progression free survival, overall survival, and recurrence) in comparison to a standard ovarian cancer therapy. This data could point out targeted therapies that will benefit advanced ovarian cancer patients.
Round 2
Reviewer 1 Report
Comments and Suggestions for Authors
The manuscript is now significantly improved, and I believe it meets the standards for publication. I have no further concerns, and I recommend the paper for acceptance in its current form.